# SAN: Hypothesizing Long-Term Synaptic Development and Neural Engram Mechanism in Scalable Model's Parameter-Efficient Fine-Tuning

Gaole Dai [* 1]   Chun-Kai Fan [* 1]   Yiming Tang [2]   Zhi Zhang [3]   Yuan Zhang [1]   Yulu Gan [4]   Qizhe Zhang [1]
Cheng-Ching Tseng [1 5]   Shanghang Zhang [✉ 1]   Tiejun Huang [✉ 1]

## Abstract

Advances in Parameter-Efficient Fine-Tuning (PEFT) bridged the performance gap with Full Fine-Tuning (FFT) through sophisticated analysis of pre-trained parameter spaces. Starting from drawing insights from Neural Engrams (NE) in Biological Neural Networks (BNNs), we establish a connection between the low-rank property observed during PEFT's parameter space shifting and neurobiological mechanisms. This observation leads to our proposed method, **Synapse** and **Neuron** (**SAN**), which decomposes and propagates scaling components from anterior feature adjusting vectors towards posterior weight matrices. Our approach is theoretically grounded in Long-Term Potentiation/Depression (LTP/D) phenomena, which govern synapse development through neurotransmitter release modulation. Extensive experiments demonstrate its effectiveness: on **vision tasks** across VTAB, FGVC, and GIC (25 datasets) using ViT, SwinT and ConvNeXt, SAN outperforms FFT up to *8.7%* and LoRA by *3.2%*; on **language tasks** using Commonsense Reasoning (8 datasets) with LLaMA models (all generations), surpassing ChatGPT up to *8.5%* and LoRA by *4.7%*; on **visual-language tasks** using Mixed Visual Instruction (7 datasets) with LLaVA models, it exceeds FFT up to *2.4%* and LoRA by *1.9%*. Our code and W&B log will be released.

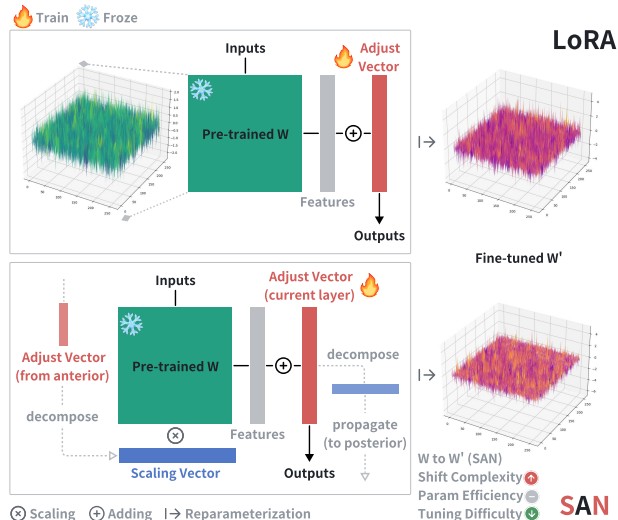

*Figure 1.* **Plug-and-Play SAN pipeline**. The core design of SAN involves extracting the scaling component by the decomposition of adjusting vectors from preceding layers and reapplying it to subsequent layers. This approach simplifies learning objectives and enhances expressiveness by providing scaling priors without introducing additional trainable parameters. SAN is compatible with various PEFT techniques, such as LoRA.

## 1. Introduction

Transfer Learning attempts to shift the pre-trained parameter space to the downstream task's optimal parameter space. Proper Full Fine-Tuning (FFT) can accomplish this task (Shuttleworth et al., 2024; Biderman et al., 2024). However, as models grow larger and tasks become more diverse, finding optimal FFT hyperparameters becomes impractical (Liu et al., 2024c). Thus, Parameter-Efficient Fine-Tuning (PEFT) becomes a better choice as it can approximately simulate FFT's functionality and find near-optimal hyperparameters through multiple training rounds at low cost (Hu et al., 2022; Liu et al., 2024b; Lian et al., 2022)

PEFT evolved from Linear Probing (LP), which approximates the shift of entire network parameters through shifting network output, to sparse tuning techniques like Bias Fine-

---

[*]Equal contribution  [1]State Key Laboratory of Multimedia Information Processing, School of Computer Science, Peking University  [2]National University of Singapore  [3]Institute for Logic, Language and Computation, University of Amsterdam  [4]Massachusetts Institute of Technology  [5]MeiTuan (This work was done when Cheng-Ching Tseng was a master student at Peking University). Correspondence to: Tiejun Huang <tjhuang@pku.edu.cn>, Shanghang Zhang <shanghang@pku.edu.cn>.

*Proceedings of the 42nd International Conference on Machine Learning, Vancouver, Canada. PMLR 267, 2025. Copyright 2025 by the author(s).*

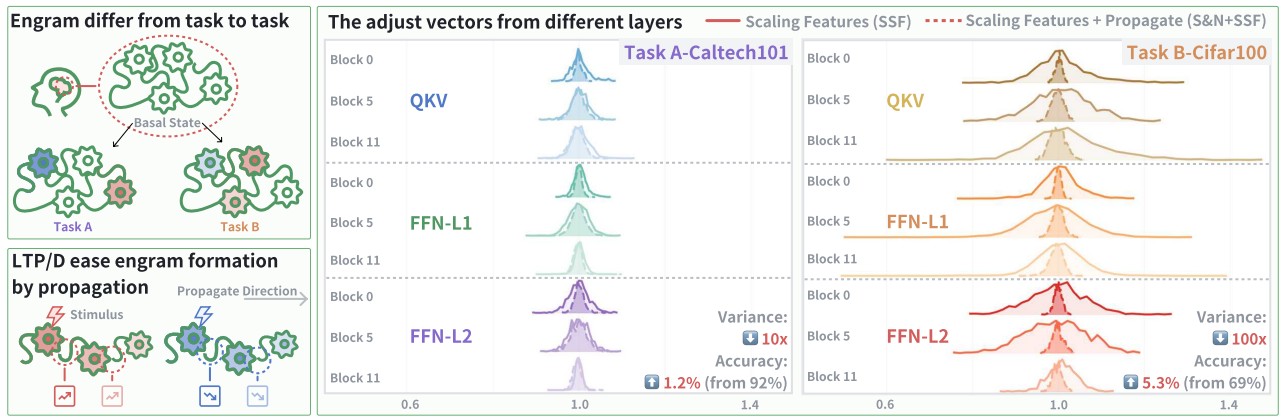

Figure 2. **Motivations and observations of decompose & propagate in SAN**. Left panels: We illustrate the concepts of Neural Engram (NE) and Long-term Potentiation/Depression (LTP/D). Right panel: we applied PEFT on pre-trained ViT-B to two VTAB subsets. This involved scaling the features using layer-wise trainable scaling vectors, akin to SSF. QKV and FFN-L1/L2 represent the Attention layer and Feed Forward Network layer 1&2, respectively. The dotted and solid lines indicate the histogram of scaling vectors from different layers with the presence or absence of SAN, respectively. Analysis of those vectors reveals significant intra-task similarities but marked inter-task differences, consistent with the principles of NE. Moreover, SAN effectively controls the variance ($\sigma^2$) of the scaling vectors, thereby allowing for more nuanced adjustments and mitigating limitations in expressiveness, which aligns with the mechanisms of LTP/D.

Tuning (BitFit) (Zaken et al., 2021) and Gradient Parameter Selection (GPS) (Zhang et al., 2024b) which more explicitly covers different network parts for parameter space shifting. Later, techniques like Low-Rank Adaptation (LoRA) (Hu et al., 2022) emerged, using additional trainable low-rank parameters to model shifting rules in the original parameter space, addressing the insufficient expressiveness of sparse tuning. Recent works like Scale and Shift (SSF) (Lian et al., 2022) and Weight-Decomposed LoRA (DoRA) (Liu et al., 2024b) systematically analyzed PEFT's shifting properties in the original parameter space, independently concluding these shifts' low-rank (approximately linear) nature. Specifically, SSF directly applies linear shifts to features, equivalent to applying linear transformation toward pre-trained parameters. The state-of-the-art performance of SSF in vision tasks unravels the minimum (i.e., linear) requirements to adapt the pre-trained model to new tasks. On the other hand, DoRA builds upon LoRA by decomposing pre-trained weight matrices into direction and magnitude components and updating the direction component by LoRA. This decomposition seems to constrain update, however, extensive experiments in scaled-up language tasks show its validity. Further demonstrated optimizing shifting strategies is more crucial than introducing more trainable parameters (e.g., increasing the rank in LoRA naively), considering the low-rank nature of both pre-trained and fine-tuned parameters.

This aligns with the *Neural Engram* (NE) (Tonegawa et al., 2015) phenomenon observed in Biological Neural Networks (BNNs), where the brain processes new knowledge by strengthening/weakening existing connections (synapses) in certain patterns, forming topologically invariant but shifted "new networks". This helps preserve energy and time costs for developing new synapses and enables rapid learning (see Figure 2, top left). Here, we considered another important BNN characteristic closely related to the formation of NE - the *Long-Term Potentiation/Depression* (LTP/D) (Malenka & Bear, 2004) phenomenon. These phenomena extend Hebb's rule (Hebb, 1949), stating that frequent strengthening/weakening of preceding related neurons directly affects subsequent neuronal development, showing a strong positive correlation (see Figure 2, bottom left). This occurs because the brain tends to simplify signal transmission mechanisms for efficiency and energy conservation. In PEFT settings, we believe this efficiency mechanism should be borrowed. Our approach comprises two parts: **(1)** we linearly decompose the shift between original and current parameter outputs (i.e. feature adjusting vectors) at the preceding layers, representing the strengthening/weakening of anterior neurons in BNN. **(2)** we reapply this decomposed scaling vector to parameters in subsequent layers, pre-scaling these pre-trained parameters (see Figure 1). By explicitly propagating these scaling vectors, similar phenomena observed in BNNs can also be identified in ANNs. In the right panel of Figure 2, we demonstrated that applying a simple feature scaling strategy in SSF for PEFT is feasible for certain tasks. However, with a more pronounced domain gap, these one-dimensional scaling vectors tend to stretch further and struggle to shift the original parameter space effectively. Our method, *Synapse and Neuron* (SAN) addresses this issue by alleviating the burden on subsequent trainable parameters. This is achieved through the decomposition and propagation of scaling vec-

tors, which reduces the complexity of remodelling shift quantities and mitigates learning difficulties with limited trainable parameters. We would further analyze those properties in Section 3.2

To summarize, our contributions can be outlined as follows:

- We investigated the low-rank characteristics of adjustment values during PEFT in ANNs and discovered a strong analogy to well-documented phenomena in BNNs, such as Neural Engram (NE) and Long-Term Potentiation/Depression (LTP/D). Our analysis of the Engram pattern in ANNs revealed significant intra-task similarities but marked inter-task differences, suggesting the potential for sharing cross-layer adjustment vectors during PEFT.

- We developed a Plug-and-Play sharing method named SAN. This approach integrates individual tuning components from different layers into a cohesive system, thereby enhancing the performance of existing PEFT methods without incurring additional trainable costs.

- Extensive experimental results demonstrate that our methods are plug-and-playable to various PEFT methods on diverse scalable pre-trained models across different tasks, including mainstream vision, language, and multi-modality benchmarks.

## 2. Related works

**Parameter-Efficient Fine-Tuning** Parameter-Efficient Fine-Tuning (PEFT) has emerged as a compelling alternative to Full Fine-Tuning (FFT), enabling the adaptation of large pre-trained models to downstream tasks with minimal additional trainable parameters. Early approaches, such as Linear Probing (LP) (Alain & Bengio, 2016), proposed adding a simple, trainable classification head to pre-trained models. While effective, this method often suffered from limited expressiveness. Subsequent advancements introduced sparse tuning techniques such as BitFit (Zaken et al., 2021), which focused on tuning only the bias terms of a model. Similarly, prompt tuning (Liu et al., 2021a; Jia et al., 2022) adjusted model inputs rather than internal parameters. These techniques leveraged the pre-trained model's latent knowledge while significantly reducing computational costs. One of the most notable PEFT advancements is the Low-Rank Adapter (LoRA) (Hu et al., 2022), which uses low-rank matrix approximations to fine-tune large models with minimal parameter overhead. LoRA's success has inspired further innovations, including quantized variations like Q-LoRA (Dettmers et al., 2024), and methods such as Scale and Shift Fine-Tuning (SSF) (Lian et al., 2022), which highlight the low-rank (approximately linear) properties of the adjustment space. DoRA (Liu et al., 2024b) further

decomposes low-rank adjustments into magnitude and direction components, revealing the nuanced dynamics of parameter shifts in fine-tuning. These methods demonstrate that strategically optimizing parameter updates, rather than naively increasing model complexity, leads to substantial improvements in transfer learning.

**Neural Engram and Long-Term Depression/Potentiation** In Biological Neural Networks (BNNs), the Neural Engram refers to the physical and functional changes in the brain associated with the encoding, storage, and retrieval of memories (Guskjolen & Cembrowski, 2023). These patterns are hypothesized to manifest as alterations in synaptic connections and activity across networks of neurons, forming topologically invariant but functionally adaptive "new networks" (Tonegawa et al., 2015). This process facilitates the efficient acquisition and integration of new information while preserving existing knowledge. The concept of the Neural Engram is particularly relevant to transfer learning, as it parallels the optimization observed in pre-trained Artificial Neural Networks (ANNs). A key mechanism underlying the Neural Engram is Long-Term Potentiation (LTP) and Long-Term Depression (LTD) (Malenka & Bear, 2004; Bear et al., 2007), which describe the strengthening and weakening of synaptic connections, respectively. These processes align with Hebbian principles (Hebb, 1949), often summarized as "neurons that fire together, wire together." LTP/D mechanisms enable efficient signal transmission by modulating synaptic weights based on activity patterns, thereby conserving energy while optimizing learning processes.

## 3. Methods

In this section, we present SAN, encompassing several preliminary methods and their corresponding adjustments.

### 3.1. Preliminaries

**LoRA & Adapter:** These PEFT methods use two low-rank learnable matrices ($\mathbf{W}^{down} \in R^{d \times r}(r \ll d)$) and $\mathbf{W}^{up} \in R^{r \times d}$) to simulate the full-rank dense layers.

$$\mathbf{y} = [\mathbf{W}^{up}\phi(\mathbf{W}^{down}\mathbf{x}^T)]^T \tag{1}$$

where $\mathbf{x}$, $\mathbf{y}$, and $\phi$ represent the inputs, outputs and linear/non-linear function, respectively.

**Scale & Shift Features (SSF):** This PEFT method applies a learnable linear transformation to each layer's output $\mathbf{y}' = \gamma \odot \mathbf{y} + \beta \in R^{n \times d}$, where $\gamma, \beta \in R^d$ are the scaling and shifting vectors, respectively, and $\odot$ is the element-wise product. The reparameterize formula is:

$$\mathbf{W}' = \gamma \odot \mathbf{W} \tag{2}$$

where $\mathbf{W}$ is the original weight of this layer.

**DoRA:** DoRA decomposes the pre-trained weight $\mathbf{W}$ into magnitude and direction components for fine-tuning:

$$\mathbf{W} = m\frac{\mathbf{V}}{||\mathbf{V}||_c} = ||\mathbf{W}||_c\frac{\mathbf{W}}{||\mathbf{W}||_c} \quad (3)$$

where $m \in R^{1 \times k}$ is the magnitude vector, $\mathbf{V} \in R^{d \times k}$ is the directional matrix, with $|| \cdot ||_c$ being the vector-wise norm across each column.

During fine-tuning, DoRA keeps $\mathbf{W}$ frozen and makes $m$ and $\mathbf{V}$ (using LoRA) trainable. After tuning, LoRA-style reparameterization is adopted:

$$\mathbf{W}' = m\frac{\mathbf{V} + \Delta\mathbf{V}}{||\mathbf{V} + \Delta\mathbf{V}||_c} \quad (4)$$

where $\Delta V$ is the directional update multiplying two low-rank matrices $\mathbf{W^{down}} \in R^{d \times r}$ and $\mathbf{W^{up}} \in R^{r \times k}$ with rank $r \ll min(d, k)$.

### 3.2. Synapse and Neuron (SAN)

**Basic Formula:** Our SAN pipeline is depicted in Figure 1. Considering the simplest combination to SSF, the scaled features $\mathbf{y}'_l$ of layer $l$ can be described as $\mathbf{y}'_l = \gamma_l \odot \mathbf{y}_l$, where $\gamma_l$ is the scaling vector (we initialize it to one) and $\mathbf{y}_l$ is the original output features.

When combined to LoRA layers, we first compute the output following standard LoRA:

$$\mathbf{y}'_l = \mathbf{y}_l + [\mathbf{W}^{up}(\mathbf{W}^{down}\mathbf{x}^T)]^T \quad (5)$$

where $[\mathbf{W}^{up}(\mathbf{W}^{down}\mathbf{x}^T)]^T$ represents the low-rank update. Then, we obtain the scaling vector by computing the element-wise ratio between $\mathbf{y}'_l$ and $\mathbf{y}_l$:

$$\gamma_l = \text{Pool}(\mathbf{y}'_l \oslash \mathbf{y}_l) \quad (6)$$

where $\oslash$ denotes element-wise division and $\text{Pool}(\cdot)$ is a dimension reduction operation. This scaling vector is then explicitly propagated to scale the pre-trained weights of subsequent layers:

$$\mathbf{W}'_{l+1} = \gamma_l \odot \mathbf{W}_{l+1} \quad (7)$$

The output goes through operations such as activation function or normalization, denoted $\sigma(\cdot)$. The output for the next layer becomes:

$$\mathbf{y}'_{l+1} = \gamma_{l+1} \odot \left(\mathbf{W}'_{l+1}\sigma(\mathbf{y}'_l) + \mathbf{b}_{l+1}\right) \quad (8)$$

Here $\mathbf{b}_{l+1}$ is the bias of the next layer.

**Explicit Propagation:** The key innovation of our SAN method lies in explicitly propagating the scaling vectors of the current layer to the parameters of the subsequent layer. This approach is motivated by a fundamental insight into the nature of transformations in neural networks: any transformation applied to current features for PEFT implicitly affects the subsequent layer's parameters.

To elaborate, consider a linear transformation $T$ applied to the features $\mathbf{y}_l$ of layer $l$, $\mathbf{y}'_l = T(\mathbf{y}_l)$. The output of the subsequent layer $l + 1$ with weight $\mathbf{W}_{l+1}$ and bias $\mathbf{b}_{l+1}$ can be expressed as, $\mathbf{y}_{l+1} = \mathbf{W}_{l+1}T(\mathbf{y}_l) + \mathbf{b}_{l+1}$. This is equivalent to:

$$\mathbf{y}_{l+1} = T(\mathbf{W}_{l+1})\mathbf{y}_l + \mathbf{b}_{l+1} = \mathbf{W}'_{l+1}\mathbf{y}_l + \mathbf{b}_{l+1} \quad (9)$$

where $\mathbf{W}'_{l+1}$ is the adjusted weight matrix.

This equivalence reveals that the transformation of features in the layer $l$ can be equalized as an adjustment to the layer's weights $l + 1$, assuming no non-linear activations are applied between these operations. Methods that apply linear transformations to features implicitly learn an adjustment matrix for the subsequent layer's weights.

While this principle is straightforward in purely linear scenarios, real-world neural networks incorporate non-linear activations and normalization layers. However, we argue that our approach remains approximately valid even in these more complex settings. This is based on two observations:

1. **Near-linear behaviour of modern activation functions:** Popular activation functions like ReLU and its variants exhibit approximately linear behaviour in their active regions. This near-linearity helps preserve scaling relationships, making it unlikely for scaling effects to be arbitrarily reversed.

2. **Optimization stability:** Modern optimization methods tend to avoid unstable paths that first reverse and then restore scaling effects. Given a fixed optimal output target, it would be inefficient and potentially destabilizing for the optimization process to first counter-adjust the model parameters and then readjust them back.

Therefore, rather than letting these scaling effects propagate implicitly, SAN makes this propagation explicit, this explicit propagation allows for more efficient parameter adaptation and provides better optimization stability.

**Expressiveness & Self-regulation:** SAN also achieve a more fine-grained adjustment by explicit propagation. Consider the reparameterization introduced in Equation. 2, SSF implies a strong assumption: for each row of the current layer's weight matrix, the scaling & shifting vectors would be the same (similar to LoRA with r=1). However, explicit

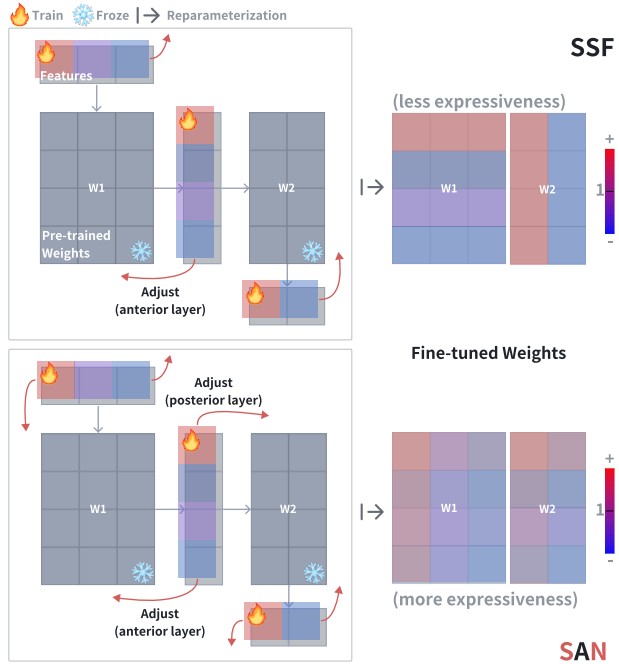

*Figure 3.* **SAN's explicit propagation mechanism.** Unlike traditional PEFT methods (e.g., SSF) that only model the shifting of the current layer (top), SAN leverages propagation to effectively adapt pre-trained parameters across layers (bottom) without introducing additional trainable parameters. The scaling vectors ($\gamma$) learned in the anterior layer are explicitly propagated to influence the pre-trained weights of posterior layers, enabling more comprehensive parameter adaptation while maintaining parameter efficiency. Gray blocks represent pre-trained frozen weights, Color blocks indicate fine-tuned trainable weights, and arrows show the adjusting direction (i.e. reparameterization direction).

propagation allows us to achieve a unique adjustment value for every parameter without incurring any extra training burden, even in such a scenario. (see Figure 3) By propagating the scaling vectors $\gamma_l$ from the current layer to the posterior layer's weight, we can overcome the strong assumption of SSF and achieve a more fine-grained adjustment. The reparameterization formula of SAN can be expressed as:

$$\mathbf{W}'_{l+1} = \gamma_{l+1} \odot (\gamma_l \odot \mathbf{W}_{l+1}) \quad (10)$$

where $\gamma_l$ is the scaling vectors of the current layer, $\gamma_{l+1}$ is the scaling vectors for the next layer.

Besides, the explicit propagation of scaling vectors in SAN introduces an implicit regularization effect to prevent overfitting. This regularization emerges from the approximate quadratic nature of the scaling vector's influence when propagated through layers. To illustrate this, let's consider a simplified two-layer linear network scenario without any activation and normalization:

$$y'_{l+1} = \gamma_{l+1} \odot ((\gamma_l \odot \mathbf{W}_{l+1})(\gamma_l \odot \mathbf{x}_{l+1}) + \mathbf{b}_{l+1}) \quad (11)$$

Rearranging this equation, we get:

$$y'_{l+1} = (\gamma_{l+1} \odot \gamma_l \odot \gamma_l \odot \mathbf{W}_{l+1})\mathbf{x}_{l+1} + \gamma_{l+1} \odot \mathbf{b}_{l+1} \quad (12)$$

The presence of $(\gamma_l)^2$ this formulation reveals a crucial property: the effect of the scaling vectors is essentially squared when propagated through layers. This quadratic influence acts as a soft constraint on the magnitude of $\gamma_l$, discouraging extreme values and promoting stability. To formalize this regularization effect, we can express it as an implicit regularization term $R(\gamma)$ added to the loss function:

$$R(\gamma) = \lambda \sum_l \|\gamma_l - 1\|^2 \quad (13)$$

where $\lambda$ is a hyperparameter controlling the strength of regularization, this regularization term penalizes large deviations $\gamma_l$ from its initial value of 1, effectively limiting the model's capacity to make extreme adaptations.

## 4. Experiments

### 4.1. Experiment Setups

This section outlines our experimental framework, including the datasets, the backbone architectures employed, and the baseline methods we compared against. For more details, please refer to the Appendix A.

**Datasets:**

- **Vision Tasks:** We evaluate on 3 major categories - **Fine-Grained Visual Classification (FGVC):** 5 specialized tasks using CUB-Birds (Van Horn et al., 2015), NA-Birds (Wah et al.), Oxford Flowers (Nilsback & Zisserman, 2008a), Stanford Dogs (Khosla et al., 2011), and Stanford Cars (Gebru et al., 2017). **Visual Task Adaptation Benchmark (VTAB-1k)** (Zhai et al., 2019): 19 diverse visual classification tasks across Natural (7 datasets), Specialized (4 datasets), and Structured (8 datasets) categories. VTAB-1k emphasizes challenging data efficiency, resulting in each dataset with a limited 1000 images for training and full test sets. **General Image Classification (GIC):** Full training and testing sets of CIFAR-100 (Krizhevsky et al., 2009) and ImageNet-1k (Deng et al., 2009).

- **Language Tasks:** Focus on Commonsense Reasoning 170k (CSR-170k) (Talmor et al., 2019) with 8 datasets: BoolQ, PIQA, SIQA, WinoGrande, HellaSwag, ARC-Easy/Challenge, and OpenBookQA (QBQA).

- **Visual-Language Tasks:** Evaluation on Mixed Visual Instruction Tuning 665k (MVIT-665K) (Liu et al., 2023a) with 7 datasets: VQA-v2, GQA, VisWiz, ScienceQA, TextVQA, POPE, and MMBench.

**Architectures:** For **vision tasks**, we use Vision Transformers (ViT) (Dosovitskiy et al., 2020), Swin Transformers (SwinT) (Liu et al., 2021b), and ConvNeXt (Liu et al., 2022). For **language tasks**, we focus on Large Language Model Meta AI (LLaMA) (Touvron et al., 2023a) family, including models from all LLaMA generations from 1-3 (LLaMA-7B, LLaMA-13B, LLaMA2-7B, and LLaMA3-8B). For **visual-language tasks**, Large Language and Vision Assistants (LLaVAs) (Liu et al., 2023a) was adopted. We use LLaVA1.5-7B (Liu et al., 2023a) and LLaVA1.5-13B (Liu et al., 2024a), to be specific.

**Baselines:** Our baselines can be divided into 3 types - **Basics:** Full Fine-Tuning (FFT), Linear Probing (LP) (Alain & Bengio, 2016), and ChatGPT with Chain of Thought (Wei et al., 2022) (in language tasks). These methods are essential to reflect the absolute performance level using other PEFT methods. **Prompt Tuning:** P-Tuning v2 (Liu et al., 2021a), Visual Prompt Tuning (VPT) (Jia et al., 2022), Neural Prompt Search (NOAH) (Zhang et al., 2024a), and Scale and Shift Features (SSF) (Lian et al., 2022). These methods derive from prompt tuning, emphasizing adjusting the inputs/features to achieve PEFT. **Model Tuning:** Bias Fine-Tuning (Bitfit) (Zaken et al., 2021), Low-Rank Adaptation (LoRA) (Hu et al., 2022), Sensitivity-Aware LoRA (SPT-LoRA) (He et al., 2023), Weight-Decomposed LoRA (DoRA) (Liu et al., 2024d), Adapter (Series/Parallel) (Zhang et al., 2020), and Adaptformer (Chen et al., 2022). These methods aim to approximate the parameter-shifting process of FFT by using limited trainable parameters, including sparse tuning techniques (e.g., Bitfit), Adapters (e.g., LoRA), and hybrid methods (e.g., SPT-LoRA).

### 4.2. Vision Task Results

Table 1 compares our proposed SAN method, integrated with SSF, against other state-of-the-art fine-tuning approaches using Vision Transformer (ViT-B) as the backbone. The results highlight the effectiveness and efficiency of SAN across a wide range of tasks and datasets. As a plug-and-play method, SAN maintains the same parameter efficiency as SSF while delivering significant improvements. On FGVC, where SSF is already considered the modern state-of-the-art (SOTA) method, SAN further enhances SSF by **0.9%**. On VTAB-1k, which has limited training set size and diversity, SSF begins to show its limitations as a trade-off for efficiency. However, SAN addresses this issue and achieves an overall improvement of **1.6%**. On GIC, where the training set size and diversity are sufficient, the performance gap between baselines is narrow; nevertheless, the benefits of adopting SAN remain evident.

We further validate SAN on more advanced backbones, such as Swin Transformers and ConvNeXt, which inherit the window mechanism and are better suited for fine-grained vision

tasks like FGVC. Our results in Figure 4 demonstrate the robustness of SAN. **When FFT surpasses SSF using newer backbones, SAN compensates for the expressiveness limitations of SSF and outperforms FFT.**

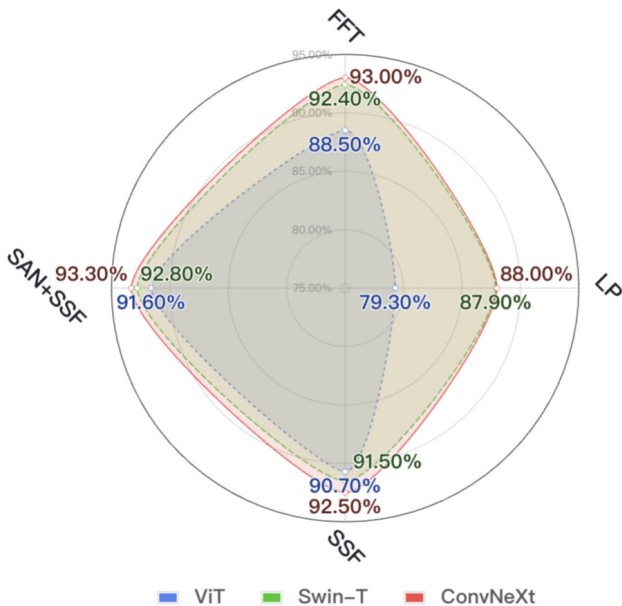

*Figure 4.* **Performance comparison on FGVC with different backbones.** Results show accuracy (%) for SAN and various baseline methods across different vision backbones

### 4.3. Language Task Results

Compared to the foundation model for pure vision tasks in Section 4.2, the importance and necessity of PEFT in large language models (LLMs) are more pronounced. Table 2 compares our proposed SAN method, integrated with LoRA and DoRA, against other PEFT approaches using LLaMA (1-3) as the backbone. The results for LLaMA-7B and 13B indicate that while prompt tuning (e.g., P-Tuning v2) is highly efficient, they exhibit limitations in terms of performance. However, increasing the trainable parameters, such as in series and parallel adapters, does not yield significant improvements. Therefore, recent advancements focus on optimizing the utilization of limited trainable parameters (e.g., DoRA derived from LoRA).

Our approach, SAN, further explores this trend. **SAN demonstrates plug-and-play capability not only for LoRA but also for prompt tuning methods like SSF. It can even be integrated with DoRA itself.** When combined with LoRA, our method outperforms DoRA, achieving accuracy gains ranging from **2.4%** (LLaMA2-7B) to **4.6%** (LLaMA3-8B). When integrated with DoRA, we still observe improvements up to **0.8%**, underscoring SAN's flexibility and broader potential.

*Table 1.* **Performance comparison using ImageNet-21k pre-trained ViT-B.** Results show accuracy (%) for SAN and various baseline methods across different vision benchmarks. We colour-coded the results red **(1st)** and blue (2nd) and the column colour reflects the baseline type. We also demonstrated a relative improvement compared with SSF.

| Baselines | Basic | | Prompt Tuning | | | | Model Tuning | | | | | | | Plug & Play |
|---|---|---|---|---|---|---|---|---|---|---|---|---|---|---|
| | FFT | LP | VPT-S | VPT-D | NOAH | SSF | Bitfit | Adapter-8 | Adapter-16 | Adaptformer | LoRA-8 | LoRA-16 | SPT-LoRA | SAN+SSF |
| ***Fine-Grained Visual Classification (FGVC)*** | | | | | | | | | | | | | | |
| Mean Param.% ↓ | 100.00 | 0.12 | 0.31 | 0.98 | 0.61 | 0.45 | 0.13 | 0.55 | 0.90 | 0.44 | 0.55 | 0.90 | 0.60 | 0.45 |
| Mean Acc.% ↑ | 88.5 | 79.3 | 84.6 | 89.1 | 85.2 | 90.7 | 88.4 | 85.5 | 85.5 | 85.1 | 86.0 | 84.8 | 90.1 | **91.6** (+0.9) |
| ***Visual Task Adaptation Benchmark (VTAB-1k)*** | | | | | | | | | | | | | | |
| Mean Param.% ↓ | 100.00 | 0.04 | 0.13 | 1.14 | 0.52 | 0.12 | 0.13 | 0.23 | 0.69 | 0.36 | 0.23 | 0.69 | 0.63 | 0.12 |
| Mean Acc.% ↑ | 69.0 | 57.6 | 67.8 | 72.0 | 75.5 | 76.1 | 65.2 | 73.9 | 74.0 | 74.8 | 74.9 | 74.9 | 76.4 | **77.7** (+1.6) |
| Natural | | | | | | | | | | | | | | |
| Mean Acc.% ↑ | 75.9 | 68.9 | 76.8 | 78.5 | 80.2 | 81.6 | 73.3 | 79.0 | 79.6 | 80.6 | 79.5 | 79.8 | 81.9 | **83.2** (+1.6) |
| Specialized | | | | | | | | | | | | | | |
| Mean Acc.% ↑ | 83.4 | 77.2 | 79.7 | 82.4 | 84.9 | 86.6 | 78.3 | 84.0 | 84.0 | 85.4 | 84.6 | 85.0 | 85.9 | **88.6** (+2.0) |
| Structure | | | | | | | | | | | | | | |
| Mean Acc.% ↑ | 47.6 | 26.8 | 47.0 | 55.0 | 61.3 | 59.9 | 44.1 | 58.5 | 58.3 | 58.5 | 60.5 | 60.2 | **61.4** | 61.3 (+1.4) |
| ***General Image Classification (GIC)*** | | | | | | | | | | | | | | |
| Mean Param.% ↓ | 100.00 | 0.48 | 0.91 | 1.42 | 0.75 | 0.69 | 0.61 | 0.80 | 1.18 | 0.72 | 0.80 | 1.18 | 0.97 | 0.69 |
| *CIFAR100* | 93.8 | 88.7 | 90.4 | 93.2 | 93.0 | 94.0 | 93.4 | 93.1 | 93.3 | 93.3 | 93.8 | 93.5 | 93.7 | **94.2** (+0.2) |
| *ImageNet-1k* | 83.6 | 82.0 | 82.1 | 82.5 | 82.6 | 83.1 | 82.7 | 83.3 | 82.7 | 83.5 | 82.6 | 82.4 | 83.7 | **83.8** (+0.7) |

### 4.4. Visual-Language Task Results

After evaluating pure vision and language tasks, we became particularly interested in multi-modal tasks due to their closer resemblance to how the human brain acquires new knowledge. Over the past year, LLaVA models have demonstrated the effectiveness of instruction tuning for visual-language tasks without relying on complex connector designs (e.g.,,, Q-former). Consequently, we chose LLaVA models as our backbone for testing.

Table 3 compares our proposed SAN method, integrated with LoRA and DoRA, against other tuning methods. It is worth noting that in the official LLaVA papers, FFT was used for transfer learning, and meticulous hyperparameter searches were conducted by the developers. However, we found that PEFT can perform comparably or even better than FFT in this scenario. Specifically, compared to FFT on the 7B model, SAN improved performance by **1.5%** and **2.4%** when integrated with LoRA and DoRA, respectively. Surprisingly, these scores are even **higher than those achieved using a larger backbone** (i.e., LLaVA1.5-13B) with FFT (67.8% on average).

### 4.5. Ablation Study: Is Topological Reasonable Propagation Necessary?

In Figure 5, we present the relative improvement achieved by incorporating SAN into LoRA with propagation to logically determined posterior layers (e.g., layer-wise application to related layer) compared to random posterior layers. The results indicate that while random propagation offers advantages in a few scenarios, it lacks robustness compared to logically determined propagation. We hypothesize that this arises from the divergence of scaling components between layers far apart in the network.

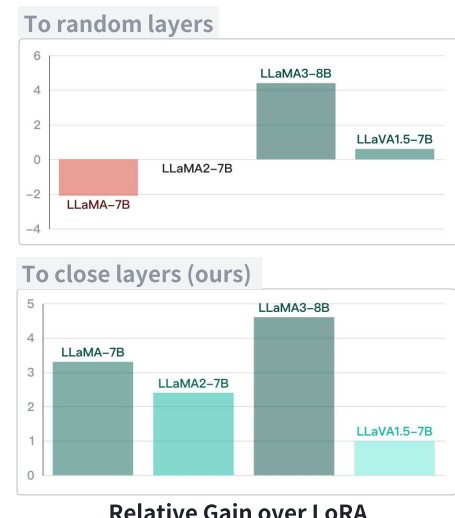

*Figure 5.* **Unstable performance when ignoring propagation order.** Blue and red bars represent positive and negative effects, respectively. Colour saturation indicates the relative gain intensity.

In our view, closely related layers are typically **locational proximity**(e.g., within the same block, next layer) or **functional proximity** (e.g., within the next block, same layer).

*Table 2.* **Performance comparison on Commonsense Reasoning using LLaMA Models.** Results show accuracy (%) for SAN and various baseline methods across CRS language benchmarks. We colour-coded the results red (1st) and blue (2nd) and the row colour reflects the baseline type (same as in Table 1). We also demonstrated a relative improvement compared with LoRA and DoRA.

| Datasets | Mean Param.% ↓ | BoolQ | PIQA | SIQA | HellaSwag | WinoGrande | ARC-E | ARC-C | QBQA | Mean Acc.% ↑ |
|---|---|---|---|---|---|---|---|---|---|---|
| ChatGPT (CoT) | - | 73.1 | 85.4 | 68.5 | 78.5 | 66.1 | 89.8 | 79.9 | 74.8 | 77.0 |
| *LLaMA-7B* | | | | | | | | | | |
| P-Tuning v2 | 0.11 | 64.3 | 76.8 | 73.9 | 42.1 | 72.1 | 72.9 | 54.0 | 60.6 | 64.6 |
| S-Adapter | 1.95 | 63.0 | 79.2 | 76.3 | 67.9 | 75.7 | 74.5 | 57.1 | 72.4 | 70.8 |
| P-Adapter | 3.54 | 67.9 | 76.4 | 78.8 | 69.8 | 78.9 | 73.7 | 57.1 | 72.4 | 71.9 |
| LoRA-32 | 0.83 | 68.9 | 80.7 | 77.4 | 78.1 | 78.8 | 77.8 | 61.3 | 74.8 | 74.7 |
| DoRA-32 | 0.84 | 69.4 | 82.4 | 78.6 | 85.3 | 81.0 | 81.9 | 66.2 | 79.2 | 78.0 |
| SAN+LoRA | 0.83 | 70.1 | 82.1 | 78.6 | 85.3 | 81.1 | 81.5 | 66.3 | 78.6 | 78.0 (+3.2) |
| SAN+DoRA | 0.84 | 71.6 | 82.6 | 79.0 | 84.9 | 82.4 | 81.0 | 66.9 | 81.8 | **78.8** (+0.8) |
| *LLaMA-13B* | | | | | | | | | | |
| P-Tuning v2 | 0.03 | 65.3 | 75.4 | 72.1 | 55.2 | 68.6 | 79.5 | 62.9 | 68.0 | 68.4 |
| S-Adapter | 1.59 | 71.8 | 83.0 | 79.2 | 88.1 | 82.4 | 82.5 | 67.3 | 81.8 | 79.5 |
| P-Adapter | 2.89 | 72.5 | 84.9 | 79.8 | 92.1 | 84.7 | 84.2 | 71.2 | 82.4 | 81.4 |
| LoRA-32 | 0.67 | 72.1 | 83.5 | 80.5 | 90.5 | 83.7 | 82.8 | 68.3 | 82.4 | 80.5 |
| DoRA-32 | 0.68 | 72.4 | 84.9 | 81.5 | 92.4 | 84.2 | 84.2 | 69.6 | 82.8 | 81.5 |
| SAN+LoRA | 0.67 | 71.9 | 84.8 | 80.0 | 91.7 | 84.5 | 84.8 | 72.1 | 83.8 | 81.7 (+1.2) |
| SAN+DoRA | 0.68 | 72.8 | 84.5 | 80.8 | 92.6 | 84.2 | 83.8 | 71.2 | 86.0 | **82.0** (+0.5) |
| *LLaMA2-7B* | | | | | | | | | | |
| LoRA-32 | 0.83 | 69.8 | 79.9 | 79.5 | 83.6 | 82.6 | 79.8 | 64.7 | 81.0 | 77.6 |
| DoRA-32 | 0.84 | 71.8 | 83.7 | 76.0 | 89.1 | 82.6 | 83.7 | 68.2 | 82.4 | 79.7 |
| SAN+LoRA | 0.83 | 72.8 | 82.6 | 79.6 | 89.8 | 81.9 | 83.2 | 70.0 | 80.4 | 80.0 (+2.4) |
| SAN+DoRA | 0.84 | 70.4 | 82.8 | 79.6 | 89.8 | 83.3 | 83.5 | 72.2 | 79.4 | **80.2** (+0.5) |
| *LLaMA3-8B* | | | | | | | | | | |
| LoRA-32 | 0.83 | 70.8 | 85.2 | 79.9 | 91.7 | 84.3 | 84.2 | 71.2 | 79.0 | 80.8 |
| DoRA-32 | 0.84 | 74.6 | 89.3 | 79.9 | 95.5 | 85.6 | 90.5 | 80.4 | 85.8 | 85.2 |
| SAN+LoRA | 0.83 | 75.0 | 88.5 | 80.0 | 95.5 | 87.5 | 90.3 | 79.6 | 86.4 | 85.4 (+4.6) |
| SAN+DoRA | 0.84 | 75.0 | 89.0 | 81.2 | 95.4 | 86.4 | 90.2 | 80.2 | 86.4 | **85.5** (+0.3) |

*Table 3.* **Performance comparison on Visual Instruction using the LLaVA Model.** Results show accuracy (%) for SAN and various baseline methods across the MVIT visual-language benchmarks. We colour-coded the results red (1st) and blue (2nd) and the row colour reflects the baseline type (same as in Table 1). We also demonstrated a relative improvement compared with LoRA and DoRA.

| Datasets | Mean Param.% ↓ | VQA-v2 | GQA | VisWiz | SQA | TVQA | POPE | MMBench | Mean Acc.% ↑ |
|---|---|---|---|---|---|---|---|---|---|
| LLaVA1.5-13B (FFT) | 100 | 80.0 | 63.3 | 56.7 | 71.6 | 48.7 | 85.9 | 68.7 | 67.8 |
| *LLaVA1.5-7B* | | | | | | | | | |
| FFT | 100 | 78.5 | 61.9 | 50.0 | 66.8 | 58.2 | 85.9 | 64.3 | 66.5 |
| LoRA-128 | 4.61 | 79.1 | 62.9 | 47.8 | 68.4 | 58.2 | 86.4 | 66.1 | 67.0 |
| DoRA-128 | 4.63 | 78.6 | 62.9 | 52.2 | 69.9 | 57.0 | 87.2 | 66.1 | 67.7 |
| SAN+LoRA | 4.61 | 79.2 | 63.4 | 51.9 | 69.8 | 58.3 | 86.8 | 66.6 | 68.0 (+1.0) |
| SAN+DoRA | 4.63 | 79.2 | 64.4 | 54.5 | 71.1 | 58.3 | 88.0 | 67.4 | **68.9** (+1.2) |

This aligns with how LTP/D forms in BNNs, where unrelated (neither locational nor functional) neurons are less likely to influence each other (Bailey et al., 2000). Please refer to Section C for more details about locational and functional proximity.

## 5. Conclusions and Future Works

Our approach Synapse and Neuron (SAN) is inspired by the Long-Term Potentiation/Depression (LTP/D) mechanism observed in Biological Neural Networks (BNNs). In BNNs, LTP/D facilitates the formation of Neural Engrams (NE). In Artificial Neural Networks (ANNs), we introduce the concept of decomposing feature adjustment values and propagating the scaling components forward to the parameters of subsequent layers. We have observed the self-regulation and plug-and-play performance-enhancing capabilities of SAN. We hypothesize that these effects are a consequence of explicit propagation. Extensive experiments validate our method, and we believe **future research should focus on discovering more NE mechanisms in scalable ANNs, particularly those related to transfer, continual, and multitask learning.**

## Acknowledgment

Gaole Dai was supported by the National Natural Science Foundation of China (Grant No. W2442028). Shanghang Zhang was supported by the National Science and Technology Major Project (Grant No. 2022ZD0117800).

## Impact Statement

This paper presents work whose goal is to advance the field of Machine Learning. There are many potential societal consequences of our work, none of which we feel must be specifically highlighted here.

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

# Appendix - SAN: Hypothesizing Long-Term Synaptic Development and Neural Engram Mechanism in Scalable Model's Parameter-Efficient Fine-Tuning

## A. Experiment Setups

**Datasets:** Our comprehensive evaluation spans across vision, language, and visual-language domains:

- **Vision Tasks:** We evaluate on 3 major categories:

  - **Fine-Grained Visual Classification (FGVC):** 5 specialized tasks using
    * CUB-Birds (Van Horn et al., 2015)
    * NA-Birds (Wah et al.)
    * Oxford Flowers (Nilsback & Zisserman, 2008a)
    * Stanford Dogs (Khosla et al., 2011)
    * Stanford Cars (Gebru et al., 2017)

    FGVC focuses on fine-grained image data classification, the definition of fine-grain in FGVC includes high-quality images and fine-class separation. All of the selected datasets have moderate numbers of training sets.

  - **Visual Task Adaptation Benchmark (VTAB-1k)** (Zhai et al., 2019): 19 diverse visual classification tasks across Natural (7 datasets):
    * CIFAR100 (Krizhevsky, 2009)
    * Caltech101 (Fei-Fei et al., 2004)
    * Oxford Flowers (Nilsback & Zisserman, 2008b)
    * Oxford Pets (Parkhi et al., 2012)
    * Describable Textures Dataset(DTD) (Cimpoi et al., 2015)
    * Sun397 (Xiao et al., 2010)
    * SVHN (Netzer et al., 2011)

    Specialized (4 datasets):
    * EuroSAT (Helber et al., 2019)
    * Patch Camelyon (Litjens et al., 2017)
    * Diabetic Retinopathy (Kaggle, 2015)
    * Resisc45 (Cheng et al., 2017)

    Structured (8 datasets):
    * Clevr Counting (Clevr-Count) (Johnson et al., 2017a)
    * Clevr Distance Prediction (Clevr-Dist) (Johnson et al., 2017b)
    * Dsprites Location Prediction (Dsprites-Loc) (Matthey et al., 2017a)
    * Dsprites Orientation Prediction (Dsprites-Ori) (Matthey et al., 2017b)
    * Smallnorb Azimuth Prediction (Smallnorb-Azi) (LeCun et al., 2004a)
    * Smallnorb Elevation Prediction (Smallnorb-Ele) (LeCun et al., 2004b)
    * Dmlab Frames (DMLab) (Beattie et al., 2016)
    * Kitti Distance Prediction (Kitti-Dist) (Geiger et al., 2012)

    VTAB-1k emphasizes challenging data efficiency, Each dataset with a limited 1000 images for training and full test sets.

  - **General Image Classification (GIC):** Full training and testing sets of
    * CIFAR100 (Krizhevsky et al., 2009)
    * ImageNet-1k (Deng et al., 2009)

- **Language Tasks:** Focus on Commonsense Reasoning 170k (CSR-170k) (Talmor et al., 2019) with 8 datasets:

  - BoolQ (Clark et al., 2019)
  - PIQA (Bisk et al., 2020)
  - SIQA (Sap et al., 2019)
  - WinoGrande (Sakaguchi et al., 2021)
  - HellaSwag (Zellers et al., 2019)
  - ARC-Easy (ARC-E) (Clark et al., 2018)
  - ARC-Challenge (ARC-C) (Clark et al., 2018)
  - OpenBookQA (OBQA) (Mihaylov et al., 2018)

- **Visual-language Tasks:** Evaluation on Mixed Visual Instruction Tuning 665k (MVIT-665K) (Liu et al., 2023a) with 7 datasets:

  - VQA-v2 (Goyal et al., 2017)
  - GQA (Hudson & Manning, 2019)
  - VisWiz (Bigham et al., 2010)
  - ScienceQA (SQA) (Lu et al., 2022)
  - TextVQA (TVQA) (Singh et al., 2019)
  - POPE (Li et al., 2023)
  - MMBench (Liu et al., 2023b)

**Architectures:**

- **vision tasks**: we use

  - Vision Transformers-Base (ViT-B) (Dosovitskiy et al., 2020)
  - Swin Transformers-Base (SwinT-B) (Liu et al., 2021b)
  - ConvNeXt-Base (Liu et al., 2022)

- **language tasks**: we focus on Large Language Model Meta AI (LLaMA) (Touvron et al., 2023a) family:

  - LLaMA-7B (Touvron et al., 2023a)
  - LLaMA-13B (Touvron et al., 2023a)
  - LLaMA2-7B (Touvron et al., 2023b)
  - LLaMA3-8B (Patterson et al., 2022)

- **visual-language tasks**: Large Language and Vision Assistants (LLaVAs) (Liu et al., 2023a) was adopted. We use:

  - LLaVA1.5-7B (Liu et al., 2023a)
  - LLaVA1.5-13B (Liu et al., 2023a)

**Baselines:** Our baselines can be divided into 3 types:

- **Basics:**

  - Full Fine-Tuning (FFT)
  - Linear Probing (LP) (Alain & Bengio, 2016)
  - ChatGPT with Chain of Thought (Wei et al., 2022) (in language tasks)

  These methods are essential to reflect the absolute performance level using other PEFT methods.

- **Prompt Tuning:**

  - P-Tuning v2 (Liu et al., 2021a)
  - Visual Prompt Tuning (VPT) (Jia et al., 2022)

  – Neural Prompt Search (NOAH) (Zhang et al., 2024a)
  – Scale and Shift Features (SSF) (Lian et al., 2022)

These methods derive from the prompt tuning technique, emphasizing shifting the inputs (or features) and achieving PEFT.

- **Model Tuning:**

  – Bias Fine-Tuning (Bitfit) (Zaken et al., 2021)
  – Low-Rank Adaptation (LoRA) (Hu et al., 2022)
  – Sensitivity-Aware LoRA (SPT-LoRA) (He et al., 2023)
  – Weight-Decomposed LoRA (DoRA) (Liu et al., 2024d)
  – Series Adapter (S-Adapter) (Zhang et al., 2020)
  – Parallel Adapter (P-Adapter) (Zhang et al., 2020)
  – Adaptformer (Chen et al., 2022)

These methods aim to approximate the parameter-shifting process of FFT by using limited trainable parameters, including sparse tuning techniques (e.g., Bitfit), Adapters (e.g., Adapter, LoRA), and hybrid methods (e.g., SPT-LoRA).

**Code-Bases:** Our implementation is based on the code of SSF `https://github.com/dongzelian/SSF` for vision tasks and DoRA `https://github.com/NVlabs/DoRA` for language and visual-language tasks. We followed most of the hyperparameter setups from these two works as a plug-and-play method. However, we searched for hyperparameters such as the learning and drop-path rate for some tasks (mainly for the VTAB-1k benchmark). More detail for hyperparameters can be found in Section B and our W&B logs

# B. Experiment Configurations

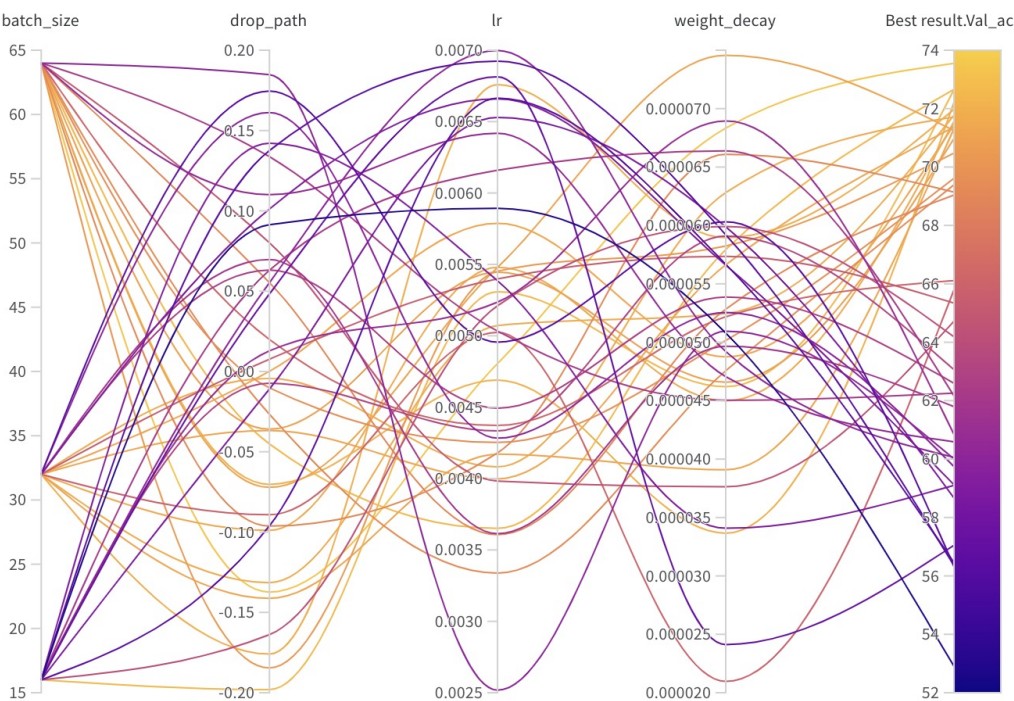

*Figure 6.* **W&B auto-sweep example on CIFAR100 with ViT-B**

We search for hyperparameters in two ways:

- **Manual Search:** For FGVC, GIC and all the language, visual-language tasks, we search for hyperparameters (mainly learning rate) manually. Notice for FGVC and GIC, each dataset is trained separately and could have different optimal hyperparameters. However, all the datasets in CRS/MVIT, are trained together and share the same hyperparameters.

- **W&B Auto-Sweep:** For VTAB-1K, since all 19 datasets need to search hyperparameters separately, we conduct W&B Auto-Sweep to narrow down the optimal range. A sample Auto-Sweep outcome is shown as Figure 6

As a result, we will list the core hyperparameter for each dataset in the upcoming paragraph.

## B.1. Vision Tasks

**Basic Informations:**

- **Optimizer:** AdamW

- **Scheduler:** Cosine Annealing with Warm-Up

- **Total Epoch Number:** 100

- **Warm-Up Epoch Number:** 10

- **Device:** NVIDIA RTX 3090 Ti x 4

**Hyperparmeters:** For more details, please refer to our W&B log, some core configurations are listed in Table 4.

*Table 4.* **Dataset-Specific hyperparameters**

| Dataset Name | Learning Rate | Drop Path Rate | Search Type |
|---|---|---|---|
| CUB-Birds | 0.01 | 0.0 | Manual |
| NA-Birds | 0.0001 | 0.1 | Manual |
| Oxford Flowers | 0.01 | 0.1 | Manual |
| Stanford Dogs | 0.00025 | 0.0 | Manual |
| Stanford Cars | 0.01 | 0.0 | Manual |
| CIFAR100 | 0.005 | 0.0 | Auto |
| Caltech101 | 0.0015 | 0.3 | Auto |
| Oxford Flowers | 0.005 | 0.0 | Manual |
| Oxford Pets | 0.005 | 0.0 | Auto |
| DTD | 0.0025 | 0.0 | Manual |
| Sun397 | 0.005 | 0.0 | Auto |
| SVHN | 0.008 | 0.1 | Auto |
| EuroSAT | 0.005 | 0.1 | Auto |
| Patch Camelyon | 0.005 | 0.0 | Auto |
| Diabetic Retinopathy | 0.005 | 0.0 | Manual |
| Resisc45 | 0.001 | 0.1 | Auto |
| Clevr-Count | 0.002 | 0.3 | Auto |
| Clevr-Dist | 0.05 | 0.1 | Auto |
| Dsprites-Loc | 0.01 | 0.2 | Manual |
| Dsprites-Ori | 0.005 | 0.15 | Auto |
| Smallnorb-Azi | 0.015 | 0.05 | Auto |
| Smallnorb-Ele | 0.005 | 0.3 | Auto |
| DMLab | 0.005 | 0.0 | Manual |
| Kitti-Dist | 0.01 | 0.0 | Manual |
| CIFAR100 | 0.001 | 0.0 | Manual |
| ImageNet-1k | 0.001 | 0.1 | Manual |

*Table 5.* **Model training hyperparameters**

| Model Name | Base Method | Learning Rate | Batch Size |
|---|---|---|---|
| LLaMA-7B | LoRA | 0.0002 | 64 |
| | DoRA | 0.0002 | 64 |
| LLaMA-13B | LoRA | 0.0002 | 32 |
| | DoRA | 0.0002 | 32 |
| LLaMA2-7B | LoRA | 0.0002 | 64 |
| | DoRA | 0.0002 | 64 |
| LLaMA3-8B | LoRA | 0.0001 | 64 |
| | DoRA | 0.0002 | 64 |
| LLaVA1.5-7B | LoRA | 0.0002 | 16 |
| | DoRA | 0.0001 | 16 |

*Table 6.* **Performance comparison on Commonsense Reasoning using LLaMA-7B Model.** Results show accuracy (%) for SAN using different setups across CRS language benchmarks. We colour-coded the row colour to reflect the baseline type (same as in Table 1).

| Datasets | Mean Param.%↓ | BoolQ | PIQA | SIQA | HellaSwag | WinoGrande | ARC-E | ARC-C | QBQA | Mean Acc.%↑ |
|---|---|---|---|---|---|---|---|---|---|---|
| ChatGPT (CoT) | - | 73.1 | 85.4 | 68.5 | 78.5 | 66.1 | 89.8 | 79.9 | 74.8 | 77.0 |
| LoRA-32 | 0.83 | 68.9 | 80.7 | 77.4 | 78.1 | 78.8 | 77.8 | 61.3 | 74.8 | 74.7 |
| Applied Layers = Q, UP, Down | 0.83 | 63.2 | 75.9 | 79.0 | 82.5 | 81.3 | 80.8 | 64.5 | 79.0 | 75.8 |
| Learning Rate = 0.0003 | 0.83 | 69.2 | 81.7 | 79.9 | 83.0 | 81.0 | 79.0 | 62.7 | 77.0 | 76.7 |
| Applied Type = Layerwise | 0,83 | 65.0 | 81.8 | 78.8 | 84.7 | 78.8 | 81.0 | 67.1 | 79.8 | 77.2 |
| *Applied Layers = Q, K, V, UP, Down / Applied Type = Blockwise / Learning Rate = 0.0002* | | | | | | | | | | |
| Adopted Setups | 0.83 | 70.1 | 82.1 | 78.6 | 85.3 | 81.1 | 81.5 | 66.3 | 78.6 | 78.0 |

## B.2. Language & Visual-Language Tasks

**Basic Informations:**

- **Optimizer:** AdamW

- **Scheduler:** Cosine Annealing with Warm-Up

- **Total Epoch Number:** 3 (language), 1 (visual-language)

- **Warm-Up Epoch Number:** 0.03

- **Device:** NVIDIA A800 x 8 x 2

**Hyperparmeters:** In this section, we present our methodology for conducting manual hyperparameter searches for language and visual-language tasks, as detailed in Table 6. Using LLaMA-7B as a case study, we observe that the number of applied layers significantly influences performance. This finding underscores the importance of adjusting more original weight regions for effective parameter-efficient fine-tuning (PEFT) in scaled-up models. In contrast, layerwise SAN and blockwise SAN exhibit minimal impact, highlighting the robustness of SAN structures. A comparison between these two variants of SAN will be provided in Section C.

# C. Variations of SAN

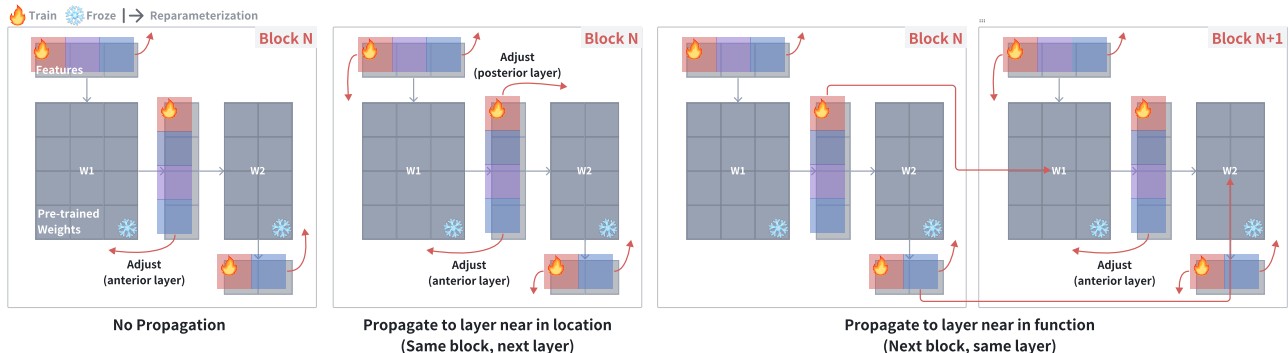

*Figure 7.* **SAN variants**

> **Recall Basic Formula of SAN**
>
> Our SAN pipeline is depicted in Figure 1. Considering the simplest combination to SSF, the scaled features $\mathbf{y}'_l$ of layer $l$ can be described as $\mathbf{y}'_l = \gamma_l \odot \mathbf{y}_l$, where $\gamma_l$ is the scaling vector (we initialize it to one) and $\mathbf{y}_l$ is the original output features.
> When combined to LoRA layers, we first compute the output following standard LoRA:
>
> $$\mathbf{y}'_l = \mathbf{y}_l + [\mathbf{W}^{up}(\mathbf{W}^{down}\mathbf{x}^T)]^T \tag{14}$$
>
> where $[\mathbf{W}^{up}(\mathbf{W}^{down}\mathbf{x}^T)]^T$ represents the low-rank update. Then, we obtain the scaling vector by computing the element-wise ratio between $\mathbf{y}'_l$ and $\mathbf{y}_l$:
>
> $$\gamma_l = \mathrm{Pool}(\mathbf{y}'_l \oslash \mathbf{y}_l) \tag{15}$$
>
> where $\oslash$ denotes element-wise division and $\mathrm{Pool}(\cdot)$ is a dimension reduction operation.

**Locational Proximity vs. Functional Proximity:** As shown in Figure 7, we propose two variants for propagating the scaling vectors: locational proximity and functional proximity. In locational proximity, as described above, the scaling vector is propagated to the subsequent layer within the same block:

$$\mathbf{W}'_{l+1} = \gamma_l \odot \mathbf{W}_{l+1} \tag{16}$$

In functional proximity, instead of propagating to the next layer, we apply the scaling vector to the corresponding layer in the next block:

$$\mathbf{W}'_{l+i} = \gamma_l \odot \mathbf{W}_{l+i} \tag{17}$$

where $i$ is the number of layers in a block, this design is motivated by our empirical observation that corresponding layers across different blocks exhibit high functional similarity. This can be attributed to the inherent functional specificity of different layer types within transformer blocks (e.g., attention layers performing QKV mapping, FFN layers handling dimension up/down scaling). As an example, in Figure 8, we observe that the perplexity remains consistently low for functionally proximate layers, even when they are several layers apart. However, this trend becomes less pronounced for extremely distant layers, which can lead to instability in the random application strategy (see Section 4.5).

While functional proximity may seem less theoretically rigorous than locational proximity, it can be approximately reduced to a locational proximity formulation if we consider each transformer block as a complex non-linear layer. Under this interpretation, reapplying scaling vectors to functionally equivalent layers becomes a natural extension of the locational proximity principle.

The final output for both variants goes through operations such as activation function or normalization, denoted $\sigma(\cdot)$:

$$\mathbf{y}'_{l+1} = \gamma_{l+1} \odot \left(\mathbf{W}'_{l+1}\sigma(\mathbf{y}'_l) + \mathbf{b}_{l+1}\right) \tag{18}$$

for locational proximity, or

$$\mathbf{y}'_{l+i} = \gamma_{l+i} \odot \left( \mathbf{W}'_{l+i} \sigma(\mathbf{y}'_{l+i-1}) + \mathbf{b}_{l+i} \right) \tag{19}$$

for functional proximity, where $b$ represents the layer bias.

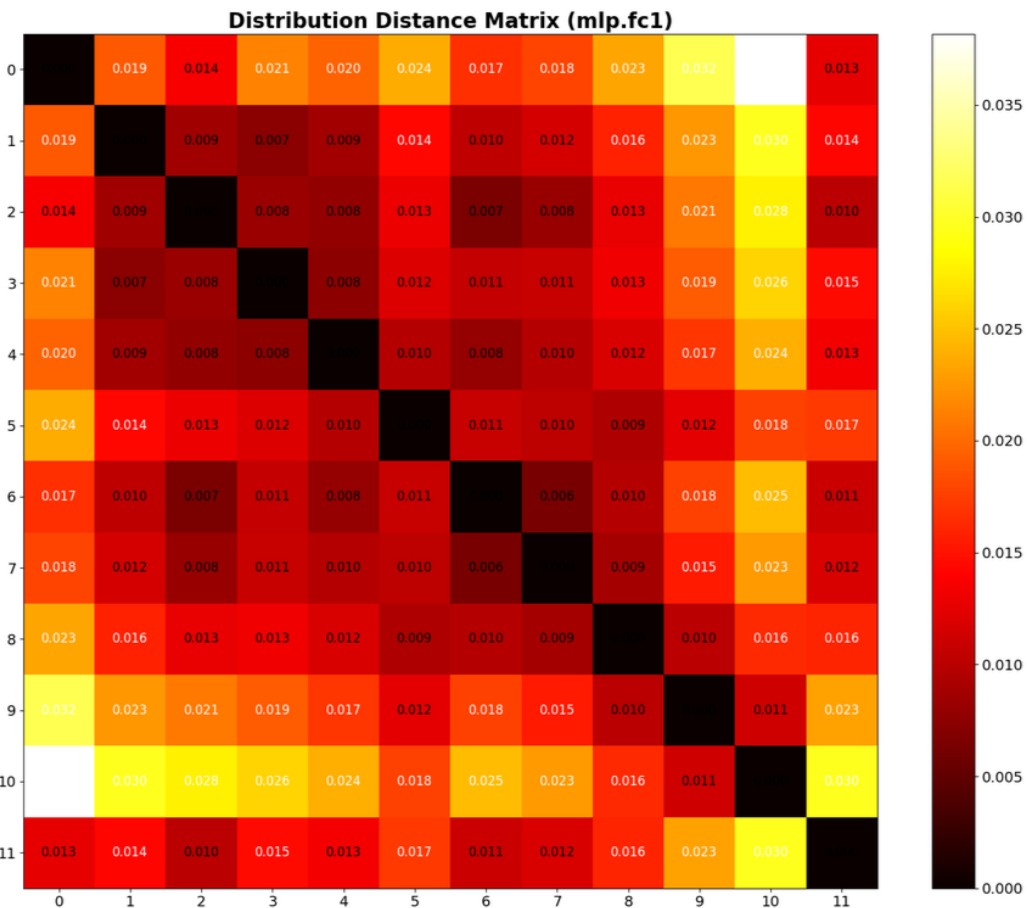

*Figure 8.* **Perplexity matrix of cross-block functional proximate layers**
We visualized the perplexity matrix of FFN first layers in ViT-B after using SSF to learn the CIFAR100 dataset.

# D. Self-regulation & Inter-task Difference/Intra-task Similarity in Neural Engram

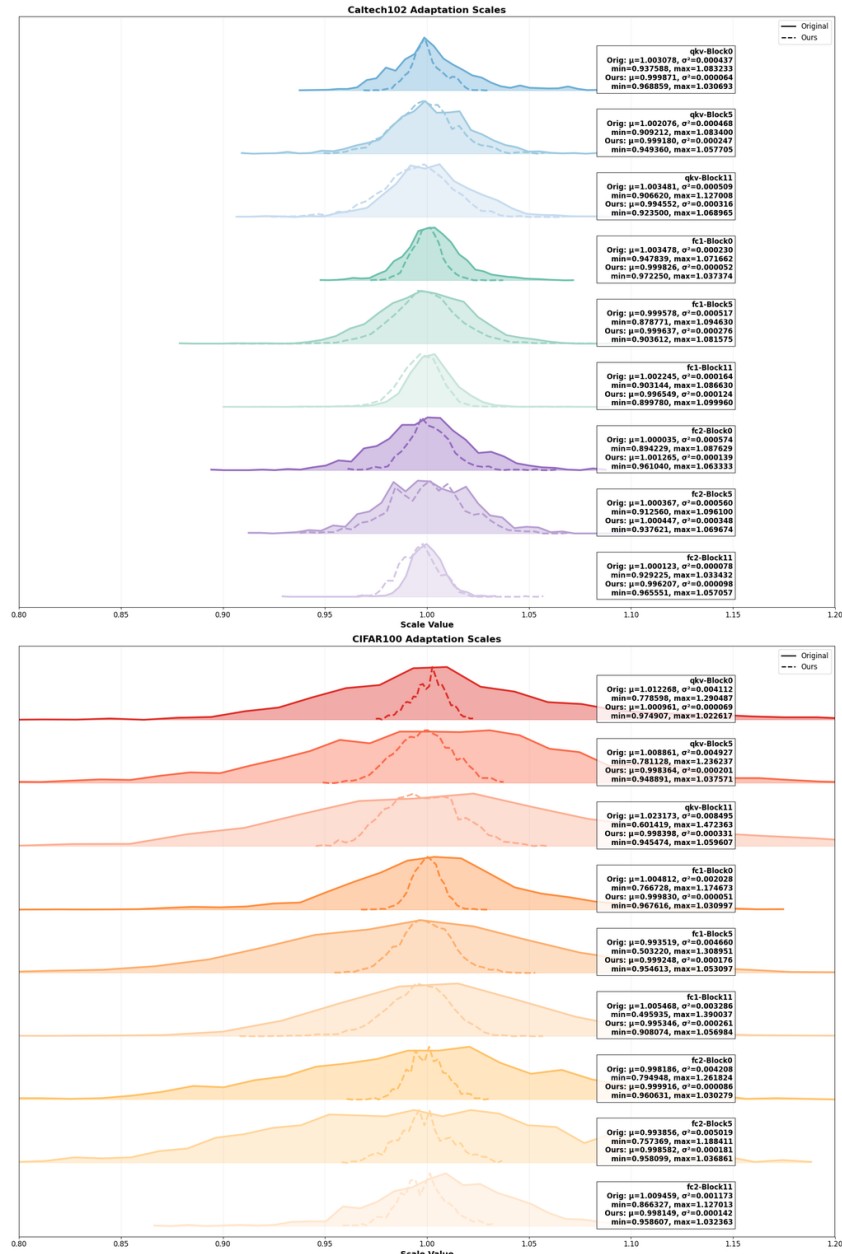

*Figure 9.* **Distribution of adjusting vectors in different task**

We have provided detailed values of the adjusting vector illustrated in Figure 2. These values substantiate our approach to self-regulation and inter-task differences/intra-task similarities during the PEFT of ANNs, drawing parallels with Neural Engrams in BNNs.

# E. Full Results of Vision Tasks

*Table 7.* **Performance comparison using Imagenet-21k pretrained ViT-B as the backbone.** Results show accuracy (%) for various fine-tuning methods across different datasets. The best results are highlighted in red (1st) and blue (2nd).

| Dataset | Basic | | Model Tuning | | Prompt Tuning | | | Plug&Play |
|---|---|---|---|---|---|---|---|---|
| | FFT | LP | Bitfit | LoRA | VPT-S | VPT-D | SSF | SAN+SSF |
| ***Fine-Grained Visual Classification (FGVC)*** | | | | | | | | |
| CUB-Brids | 87.3 | 85.3 | 87.1 | 86.7 | 86.7 | 88.5 | 89.5 | **90.6** (+1.1) |
| NA-Brids | 82.7 | 75.9 | 84.3 | 78.8 | 78.8 | 84.2 | 85.7 | **86.3** (+0.6) |
| Oxford Flowers | 98.8 | 97.9 | 98.5 | 98.4 | 98.4 | 99.0 | 99.6 | **99.7** (+0.1) |
| Stanford Dogs | 89.4 | 86.2 | 89.8 | 90.7 | 90.7 | 90.2 | 89.6 | **91.1** (+0.5) |
| Stanford Cars | 84.5 | 51.3 | 68.6 | 68.7 | 68.7 | 83.6 | 89.2 | **90.4** (+1.2) |
| ***Visual Task Adaptation Benchmark (VTAB-1k)*** | | | | | | | | |
| CIFAR100 | 68.9 | 63.4 | 72.8 | 68.1 | 77.7 | **78.8** | 69.0 | 74.3 (+5.3) |
| Caltech101 | 87.7 | 85.0 | 87.0 | 91.4 | 86.9 | 90.8 | 92.6 | **93.8** (+1.2) |
| Oxford Flowers | 97.9 | 97.2 | 97.5 | 99.0 | 97.5 | 98.0 | 99.4 | **99.7** (+0.3) |
| Oxford Pets | 86.9 | 86.3 | 85.3 | 90.5 | 87.3 | 88.3 | 91.8 | **93.0** (+1.2) |
| DTD | 64.3 | 64.1 | 59.2 | 69.8 | 62.6 | 65.8 | 75.1 | **76.4** (+1.3) |
| Sun397 | 38.8 | 51.0 | 51.4 | 53.1 | 51.2 | 49.6 | 52.9 | **53.4** (+0.5) |
| SVHN | 87.4 | 36.6 | 60.0 | 86.4 | 74.5 | 78.1 | 90.2 | **91.8** (+1.6) |
| EuroSAT | 95.7 | 87.5 | 91.6 | 95.8 | 92.0 | 96.1 | 95.9 | **97.7** (+1.8) |
| Patch Camelyon | 78.9 | 78.5 | 78.7 | 85.1 | 78.2 | 81.8 | 87.4 | **88.1** (+0.7) |
| Diabetic Retinopathy | 73.9 | 74.0 | 69.8 | 74.2 | 72.9 | 68.4 | 75.5 | **78.1** (+2.6) |
| Resisc45 | 84.2 | 68.6 | 73.0 | 84.7 | 75.6 | 83.4 | 87.4 | **90.6** (+3.2) |
| Clevr-Count | 56.3 | 34.3 | 61.5 | **83.0** | 50.5 | 68.5 | 75.9 | 82.4 (+6.5) |
| Clevr-Dist | 58.6 | 30.6 | 55.6 | **66.9** | 58.6 | 60.0 | 62.3 | 61.4 (-0.9) |
| DSprites-Loc | 57.5 | 12.5 | 66.6 | 80.2 | 68.7 | 73.6 | 77.3 | **81.7** (+4.4) |
| DSprites-Ori | 46.7 | 20.0 | 40.0 | 46.6 | 36.1 | 47.9 | 54.9 | **55.2** (+0.3) |
| Smallnorb-Azi | 25.7 | 9.6 | 15.7 | **32.2** | 20.2 | 32.9 | 29.5 | 30.3 (+0.8) |
| Smallnorb-Ele | 29.1 | 19.2 | 25.1 | **41.1** | 34.1 | 37.8 | 37.9 | 40.4 (+2.5) |
| DMLab | 41.7 | 33.2 | 32.4 | 50.4 | 40.5 | 46.5 | 53.3 | **54.5** (+1.2) |
| KITTI/distance | 65.5 | 55.4 | 55.9 | 81.4 | 67.1 | 72.8 | 80.6 | **82.1** (+1.5) |
| ***General Image Classification (GIC)*** | | | | | | | | |
| CIFAR100 | 93.8 | 88.7 | 93.4 | 93.8 | 90.4 | 93.2 | 94.0 | **94.2** (+0.2) |
| ImageNet-1k | 83.6 | 82.0 | 82.7 | 82.6 | 82.1 | 82.5 | 83.1 | **83.8** (+0.7) |

Due to the page limitation of the main text, we were unable to include the comprehensive results for all 25 datasets in our vision task. However, the complete results are provided in Table 7. These results demonstrate a consistent improvement in performance following the implementation of SAN. Furthermore, in certain datasets, we observed a significant increase in accuracy (e.g., +6.5% in CLEVR-Count).

# F. Rank Ablation using LoRA-like Base Method

*Table 8.* **Performance comparison on Commonsense Reasoning using LLaMA-7B Model.** Results show accuracy (%) for SAN using different ranks across CRS language benchmarks. We colour-coded the row colour to reflect the baseline type (same as in Table 1). The best results are highlighted in red (1st) and blue (2nd).

| Datasets | Mean Param.% ↓ | BoolQ | PIQA | SIQA | HellaSwag | WinoGrande | ARC-E | ARC-C | QBQA | Mean Acc.% ↑ |
|---|---|---|---|---|---|---|---|---|---|---|
| ChatGPT (CoT) | - | 73.1 | 85.4 | 68.5 | 78.5 | 66.1 | 89.8 | 79.9 | 74.8 | 77.0 |
| LoRA-16 | 0.42 | 69.9 | 77.8 | 75.1 | 72.1 | 55.8 | 77.1 | 62.2 | 78.0 | 70.9 |
| DoRA-16 | 0.43 | 70.0 | 82.6 | 79.7 | 83.2 | 80.6 | 80.6 | 65.4 | 77.6 | 77.5 |
| SAN+LoRA-16 | 0.42 | 69.3 | 82.3 | 76.8 | 86.3 | 80.2 | 81.0 | 63.1 | 80.0 | 77.4 (+6.5) |
| SAN+DoRA-16 | 0.43 | 68.5 | 81.6 | 78.9 | 87.2 | 81.1 | 81.4 | 64.6 | 79.6 | **77.9** (+0.4) |
| LoRA-32 | 0.83 | 68.9 | 80.7 | 77.4 | 78.1 | 78.8 | 77.8 | 61.3 | 74.8 | 74.7 |
| DoRA-32 | 0.84 | 69.4 | 82.4 | 78.6 | 85.3 | 81.0 | 81.9 | 66.2 | 79.2 | 78.0 |
| SAN+LoRA-32 | 0.83 | 70.1 | 82.1 | 78.6 | 85.3 | 81.1 | 81.5 | 66.3 | 78.6 | 78.0 (+3.2) |
| SAN+DoRA-32 | 0.84 | 71.6 | 82.6 | 79.0 | 84.9 | 82.4 | 81.0 | 66.9 | 81.8 | **78.8** (+0.8) |

Although DoRA and SAN employ distinct strategies and are fully compatible with each other, similar levels of robustness were observed in Table 8 when using both methods. We hypothesize that this is due to the decomposition process simplifying the deviation requirements for the limited number of trainable parameters. Furthermore, SAN can be integrated into a wider array of tuning methods, including those that do not resemble LoRA-like approaches, which enhances its flexibility.

## G. World Model Video Generation Task

We fine-tuned Dynamicrafter (Xing et al., 2023), an open-domain image-to-video diffusion model, into a robotic manipulation-specific model by following the configurations outlined in Embodied World Model for Future Video Prediction (EVA) (Chi et al., 2024). The dataset used for this fine-tuning process was RoboVQA (Sermanet et al., 2023).

The primary objective of integrating world knowledge into the video generation model is to predict plausible future actions for specific target objects. This necessitates that the model possess robust capabilities in both maintaining consistency and adhering to instructions.

As shown in Figure 10, SAN+LoRA exhibited superior performance compared to LoRA alone. Specifically, LoRA frequently failed to generate any meaningful action, resulting in static videos that remained unchanged from the initial frame. In the limited instances where dynamic results were observed (e.g., the first case), LoRA generated the movement of the robot arm but failed to close the drawer, indicating insufficient acquisition of world knowledge. While FFT mitigated some of these issues, significant problems persisted, preventing strict success criteria from being met. In contrast, SAN demonstrated impressive results. For instance, it successfully closed the middle drawer while leaving the top drawer open as intended. Additionally, SAN not only picked up the green bag but also accurately predicted its cover and displayed it to the camera. Furthermore, SAN exhibited strong world knowledge in determining relative positions, making it the only model to achieve strict success by moving the sponge near the phone.

| Inputs | | Outputs (World Knowledge Key Frame) | | |
|---|---|---|---|---|
| **Prompt** | **Initial Frame** | **FFT** | **LoRA** | **SAN+LoRA** |
| close the middle drawer | | partial success | failure | success |
| pick up the green chip bag | | failure | failure | success |
| move the sponge to blueberry phone | | partial success | failure | success |
| put the glass cup down | | failure | failure | success |

ABC action text  ABC target text  ↓ target action  ✓ success  ✗ failure  ⚑ partial success

*Figure 10.* **Next-frame prediction with world knowledge** The outcomes of success, failure, and partial success are evaluated based on four criteria: (1) action completeness, (2) action correctness, (3) target accuracy, and (4) frame consistency. For the generation process, the inputs consist solely of a text prompt and an initial frame. Target actions are provided here for reference to enhance understanding.

