# OpenReview forum: "SAN: Hypothesizing Long-Term Synaptic Development and Neural Engram Mechanism in Scalable Model's Parameter-Efficient Fine-Tuning"
_ICML.cc/2025/Conference — ICML 2025 poster_

### Official Review · Reviewer_1yzx · 2025-03-06

**Overall Recommendation:** 3

**Summary:**

This work seeks to further advance Parameter-Efficient Fine-Tuning (PEFT) techniques, which reduce memory usage and computational cost compared to full fine-tuning. It draws insights from the Neural Engram (NE) phenomenon, where the brain processes new knowledge by strengthening or weakening existing connections, which helps to preserve energy and reduce the time costs of developing new synapses and enables rapid learning. To do this, the authors propose SAN, with the key innovation lying in explicitly propagating the scaling vectors of the current layer to the parameters of the subsequent layer, mimicking LTP/LTD. In a way, SAN explicitly propagates the layer transformation effect through a scaling approximation. This explicit propagation allows for more efficient parameter adaptation and provides certain implicit regularizations that discourage extreme values and promote stability.

## update after rebuttal
I maintain my assessment, a weak accept, as I see no major weaknesses left to the best of my knowledge

**Claims And Evidence:**

-	[Strength] The authors performed extensive empirical experiments across well-known vision tasks, language tasks, and visual-language tasks using SoTA architectures for the task. By doing so, the authors demonstrated the effectiveness of incorporating SAN into various PEFT methods (e.g., LoRA and DoRA).
-	[Strength] The authors clearly explained how SAN, with the same parameter efficiency as SSF, is more expressive than SSF.
-	[Strength] The authors clearly articulated the connection of SAN to LTP/LTD and provided further ablation studies.
-	[Weakness] While the authors explained how the principle of SAN, assuming linear transformation, can still remain approximately valid in more complex settings, it would be helpful to provide more discussion on the limitations of such an approximation and potential failure modes.
-	[Weakness] Code and hyperparameter details are stated to be released in the future, so I cannot be certain about the reproducibility of this work. That said, the authors did provide basic information on hyperparameter tuning in Appendix B.

**Essential References Not Discussed:**

None to my knowledge

**Experimental Designs Or Analyses:**

I checked all tables and figures in the main text

**Methods And Evaluation Criteria:**

See above

**Other Comments Or Suggestions:**

Some typos: missing period at the end of abstract; “(see Figure 3)” should be in front of the period on page 4 under “Expressiveness & self-regulation”.

**Other Strengths And Weaknesses:**

n/a

**Questions For Authors:**

-	By explicitly propagating the scaling vectors, would it introduce other implicit regularizations besides the one that the author mentioned, such as effects associated with reducing depth?
-	Can you provide a complexity analysis?

**Relation To Broader Scientific Literature:**

The paper is well-positioned with a clear objective

**Theoretical Claims:**

The results are mainly empirical

---

> ### Author Rebuttal · Authors · 2025-03-29
>
> ## Response to Reviewer 1yzx
> Thank you for your thorough review and feedback. Your positive comments (such as acknowledging our clearly explained SAN and its connection to LTP/LTD) and suggestions have greatly helped us improve our paper. To address your concerns, we have carefully read your feedback and provided point-by-point responses below:
> 1.  Minor typos
>
>     a.  missing period at the end of the abstract
>
>     b.  “(see Figure 3)” should be in front of the period on page 4 under “Expressiveness & self-regulation”.
>
> 2.  Provide more discussion on the limitations of the linear transformation approximation and potential failure modes.
>
> 3.  By explicitly propagating the scaling vectors, would it introduce other implicit regularizations besides the self-regularization mentioned, such as effects associated with reducing depth?
>
> 4.  Provide a complexity analysis and more hyperparameter details.
>
> ## Point-by-Point Response:
> 1.  ### Regarding Typos
> Thank you for your meticulous review. We will correct these in the revised version.
>
> 2.  ### Regarding potential failure modes
> We envision SAN as a plug-and-play method adaptable to current and future SOTA PEFT approaches across domains. This implies that SAN's failure mode mainly depends on the selected base method.
>
> To stress-test pure linear transformations in large models, we evaluated SSF and SSF+SAN on LLaMA fine-tuning. The results (see Table below) indicate that for large models (7B+ parameters), pure linear transformations of features cannot effectively align pre-trained parameters to downstream tasks. This is reasonable, as although such methods achieve excellent results on vision foundation models (which typically have only hundreds of millions of parameters) by providing orthogonal fine-tuning-like **[1]** properties that preserve hypersphere energy and prevent catastrophic forgetting, their expressive power becomes constrained in larger language/multimodality models.
> | Method |  P-Tuning V2 | SSF | SSF+SAN | LoRA | LoRA+SAN |
> | ---- | ---- | ---- | ---- | ---- | ---- |
> | Common Sense Reasoning | 64.60% | 52.60% | 61.10% | 74.70% | 78.00% |
>
> 3.  ### Regarding other implicit regularizations
> Self-regularization emerges directly from the implicit $\gamma^2$ term during adjustment, which guides scaling factors to reduce extreme values and control variance. Besides, we haven't observed significant regularization effects beyond self-regularization.
>
> To our knowledge, the depth-reducing phenomenon you mentioned appears in two types of literature: structural pruning (particularly layer pruning) like Layer Folding **[2]**, which aim to improve model efficiency; and layer redundancy studies like stochastic depth **[3]**, which prevent overfitting through random depth dropout during training. Our method doesn't directly relate to these works.
>
> 4.  ### Regarding complexity and hyperparameters
> We have provided basic hyperparameters in the paper, and more detailed parameter settings can be found in our supplementary code. As mentioned in Appendix B, we've recorded thousands of experimental setups and results on WandB, which we will make public.
>
> For algorithm complexity, SAN, as a plug-and-play method, adds no additional parameters during training. The computational cost only appears in the decompose & propagate operations. Specifically:
> -   The decomposition cost doesn't involve matrix multiplication (using average pooling or direct reuse), making the additional computation negligible.
> -   For propagation, we multiply the decomposed $1×d$ vector with a $d×k$ posterior weight matrix. The complexity for a single propagation is $O(d×k)$, where $d$ is the input feature dimension and $k$ is the output dimension. If we have multiple layer groups $(1,2,...N)$ that need propagation, the total computational complexity would be the sum of each group's complexity: $O(∑(d_i×k_i)$ for $i$ from $1$ to $N$.
>
> This minimal computational overhead makes SAN an efficient enhancement to existing PEFT methods. Practically speaking, throughout our experiment, we also did not see a significant slowdown by adding SAN.
>
> ## Reference
> **[1] Qiu, Z., Liu, W., Feng, H., Xue, Y., Feng, Y., Liu, Z., ... & Schölkopf, B. (2023). Controlling text-to-image diffusion by orthogonal finetuning.** _**Advances in Neural Information Processing Systems**_**,** _**36**_**, 79320-79362.**
>
> **[2] Dror, A. B., Zehngut, N., Raviv, A., Artyomov, E., Vitek, R., & Jevnisek, R. (2021). Layer folding: Neural network depth reduction using activation linearization. arXiv preprint arXiv:2106.09309.**
>
> **[3] Huang, G., Sun, Y., Liu, Z., Sedra, D., & Weinberger, K. Q. (2016). Deep networks with stochastic depth. In** _**Computer Vision** **ECCV 2016: 14th European Conference, Amsterdam, The Netherlands, October 11–14, 2016, Proceedings, Part**_ _**IV**_ _**14**_ **(pp. 646-661). Springer International Publishing.**

---

> > ### Comment · Reviewer_1yzx · 2025-04-05
> >
> > Thank you for your clarifications. I would like to maintain my score for now.

---

> > > ### Author Response · Authors · 2025-04-08
> > >
> > > Thank you for taking the time to provide your rebuttal comment. We are truly pleased to see your acknowledgment and hope that we have adequately addressed your concerns. Your valuable suggestion has greatly helped us enhance our manuscript.

---

### Official Review · Reviewer_bAk4 · 2025-03-10

**Overall Recommendation:** 3

**Summary:**

This paper proposes a fine tuning method based on ideas from the Neural Engram and LTPD literature which gives an alternative method to the popular LORA and DORA updates that have recently been employed for large models. The update consists computing scale vectors $\gamma_\ell$ at each hidden layer $\ell$ of the network from the pooled ratios of activity patterns $y'/y$ of the perturbed $y'$ and unperturbed neural activations $y$.  The authors show that their fine tuning method improves full fine tuning and LORA on a number of vision and language benchmarks. Part of the key idea is to maintain stable subnetworks (engrams) in the model which do not change in topology during fine tuning but rather change in precise weight scales for existing nonzero connections. The authors show that organizing updates by correct layer order is important to reap the full benefits of the method.

**Claims And Evidence:**

First, on the empirical side, this paper provides a very large set of fine tuning experiments in many modalities including vision, commonsense reasoning, and multimodal visual language tasks. I appreciate the breadth of experiments and the efforts the authors went to to benchmark their method. The fine tuning method shows promise on each of these provided experiments.

However, there are a number of claims made about both (1) the motivation and design of the algorithm and (2) theoretical claims about how /why it works that are not clearly tested. I was hoping to see some numerical evidence that their fine tuning method preserved or encouraged formations of new non-overlapping neural engrams which was the motivation of the algorithm. However, I could not find any experimental evidence that this engram formation actually occurs in their finetuning. Second, it is unclear why the pooling and elementwise division to form $\gamma_\ell$ should give rise to engrams. Lastly, the claim that $\gamma_\ell$ should be close to unity.  The authors write "This quadratic influence acts as a soft constraint on the magnitude of γl, discouraging extreme values and promoting stability." Where is this shown or argued for? Partly, I think these claims are important scientifically since the authors are claiming that this algorithm has something to do with engrams and LTP/D and that these are key insights that enabled the improvement in performance.

**Essential References Not Discussed:**

Not to my knowledge.

**Experimental Designs Or Analyses:**

I think the empirical design is valid provided that parameter counts are controlled across baselines (see questions below).

**Methods And Evaluation Criteria:**

Yes, the benchmark datasets and evaluation criteria make sense.

**Other Comments Or Suggestions:**

1. Abstract final sentence missing final punctuation  “.”
2. There are a lot of acronyms like PEFT, SSF, LORA, DORA, LP, SAN, FFT, NE, BNN, LTP. I understand this is a stylistic choice, but sometimes it is hard for the reader to keep straight. Please consider using the full name for some of the less frequently used acronyms.
Line 92 “Further demonstrated … ” should be “The authors of [cite] further demonstrated …”
3. The shapes of the matrices reported do not make sense. Notationally, the $W_{down}$ in equation 1 should be an element of $\mathbb R^{r \times d}$ and $W_{up}$ should be $\mathbb R^{d \times r}$
4. More explanation is needed around equation 1, what do x, y represent. Is there one of these for each hidden layer of the network, etc?
5. Equation 2 shapes don’t make sense. $W$ should be $\mathbb R^{d \times d}$ and $\gamma$ is a vector of size $\mathbb R^{d}$. Do the authors mean  $(\gamma 1^\top ) \odot W$ where 1 is the vector of all ones?
In Equation 9, the T( ) function is overloaded. Suppose that T(y) = A y for matrix A. Then the transformation is  W T(y) = W A y = (W A) y  = T_2( W)  y where T2(W) = W A.
6. In line 184 the authors introduce two assumptions near linear behavior + optimization stability. Under what conditions do they expect these to hold? Also in what sense is ReLU near linear? For a single input, the function is locally linear but over the space of all inputs, it is a complicated function.
7. Is equation 6 computed per-example? Does each data point have its own $\gamma_\ell$?
In Figure 3, the top of the image says “froze” but should say “frozen”
Equation 10 should (probably, if I understand correctly) read as $W’ =  ( \gamma_{\ell+1} \gamma_\ell^\top ) \odot W$.
8. The argument below eq 12 is unclear to me. Why does the update stabilize or become implicitly regularized to be near 1? Please provide a derivation or argument in the Appendix.
9. Line 369 “Topological reasonable” -> “Topologically Reasonable”

**Other Strengths And Weaknesses:**

This paper has interesting ideas and good experiments. However, I found it challenging to read / parse / understand partly due to the abundance of acronyms and partly due to poor mathematical notation and some missing explanations (see comments/questions below).

**Questions For Authors:**

1. Is there a biological motivation for equation 6? Is there any prior work that shows that this kind of plasticity rule encourages formation of engrams?
2. In the baselines, are all finetuning methods employing equal numbers of parameters or similar compute? This is important to make comparisons across methods.

**Relation To Broader Scientific Literature:**

The authors claim that their algorithm is inspired by neural engrams, which are an idea from computational neuroscience where combinatorial subsets of neurons support representations of new memories that do not interfere with previously acquired knowledge. If they could more clearly make this connection and show how their algorithm generates these engrams, I think this work would be not only interesting to finetuning researchers but also neuroscientists interested in memory formation and continual learning.

**Theoretical Claims:**

There are some issues with mathematical notation that make it difficult to actually understand some of the definitions and mathematical claims. However, I think I understand the intended interpretation of the algorithm and have read through their theoretical results.

The authors invoke two assumptions which they do not clearly justify or reference to an appropriate article. Specifically, the allude to "near-linear behaviour of modern activation functions" and also claim "Modern optimization methods tend to avoid unstable paths that first reverse and then restore scaling effects." What evidence do the authors have for these claims?

---

> ### Author Rebuttal · Authors · 2025-03-29
>
> ## Response to Reviewer bAk4
> Thank you for your thorough review and feedback. Your positive comments (such as acknowledging our solid experiment and idea) and suggestions have greatly helped us improve our paper. To address your concerns, we have carefully read your feedback and provided point-by-point responses below:
> 1. Challenging to read / parse / understand partly due to the abundance of acronyms and partly due to poor mathematical notation and some missing explanations:
>
>     a. Minor Typos and overwhelming acronyms
>
>     b. The $W_{down}$ in Equation 1 should be $\mathbb{R}^{r\times d}$ and $W_{up}$ should be $\mathbb{R}^{d\times r}$. What do $x,y$ represent?
>
>     c. $W$ in Equation 2 should be $\mathbb{R}^{d\times d}$ and $γ$ is a vector of size  $\mathbb{R}^{1\times d}$. Do the authors mean $(γ1^⊤)⊙W$, where 1 is the vector of all ones? In Equation 9, the $T(\cdot)$ function is overloaded.
>
>     d. Why does the update stabilize or become implicitly regularized to be near 1 in Equation 12? Under what conditions do the assumptions in line 184 expect to hold?
>
> 2. Is there biological motivation for equation 6? Is it computed per example??
> 3. In the baselines, are all fine-tuning methods employing equal numbers of parameters or similar compute?
>
> ## Point-by-Point Response:
> 1. ### Regarding mathematical notation and explanations
>     **a:** Thank you for your meticulous review. We will correct these points in the revised version.
>
>     **b:**  In Equation 1, we denote  $r\ll d$, where $r$ is the rank of the LoRA/Adapter. Accordingly, $W_{down}$ is defined as $\mathbb{R}^{d\times r}$ to compress the input $x$ from dimension $d$ to $r$, and $W_{up}$ is defined as $\mathbb{R}^{r\times d}$ to recover the original dimension. As noted in [Line 143-144], $x$, $y$, and $\theta$ represent the inputs, outputs, and the linear/non-linear function of the LoRA/Adapter, respectively.
>
>     **c:** $W$ can also be shaped as $\mathbb{R}^{d\times k}$ (e.g., the FFN in transformers), as long as the scaling vector $γ$ of size  $\mathbb{R}^{1\times d}$ is available for per-row scaling.  We did not locate the expression  $(γ1^⊤)⊙W$ mentioned in your comment. We assume your concern may relate to whether $γ$ is an all-one vector. Initially, $γ$ is indeed an all-one vector, but after training, it diverges (i.e., $γ$ is trainable or generated by trainable modules). For the function $T(\cdot)$, defined as $T(y_{l})=\gamma_{l}\odot y_{l}$, this represents element-wise scaling of $y_{l}$ prior to feeding it into layer $l+1$ layer. Specifically, we have: $y_{l+1}=W_{l+1}T(y_{l})+b_{l+1}=W_{l+1}\gamma_{l}\odot y_{l}+b_{l+1}=\gamma_{l}\odot W_{l+1}y_{l}+b_{l+1}=T(W_{l+1})y_{l}+b_{l+1}$, which is automatically managed by PyTorch’s broadcast mechanism.
>
>     **d:** The implicit regularization effect arises from the explicit propagation, which introduces a quadratic effect ($(\gamma_l)^2$), which discourages extreme values and thus stabilizes the updates. This stabilization prevents overfitting by controlling the magnitude of divergence from the initial value 1. In a linear scenario, scaling the output is equivalent to scaling the weights in the next layer. Although ReLUs are not globally linear, they operate nearly linearly in their active regions (when the output is non-zero).
>
> 4. ### Regarding biological motivation for Equation 6
> Equation 6 reduces both the batch and token dimensions (related to the size of data) while preserving the embedding dimension (related to the content of data), resulting in a 1D vector that can be used for scaling weights. It aligns with the principles of synaptic plasticity. Network signals are data-driven, and synaptic development adapts to these signals. For instance, high-frequency stimulation strengthens synaptic connections between neurons, and such strengthened connections maintain over a range of time, forming a specialized engram.
> Furthermore, while methods like SSF and DoRA apply scaling factors at the neuron level (using the same scaling for all parameters in each row of the weight), SAN operates at the synapse level. By introducing a propagation mechanism, we use the previous layer’s scaling as additional per-column scaling for the current layer. Resulting in more complex engram patterns, wherein each synapse within a neuron can have its scaling factor (see Figure 3). This design better aligns with recent neuroscience findings on neuronal engrams, such as the linking, buffer, and feature neuron engrams discussed by Choucry et al. (2024) **[1]**.
>
> 3. ### Regarding trainable parameters of baselines
> SAN itself doesn't introduce additional trainable parameters; our experimental parameter settings follow those established in SSF, SPT-LoRA, and DoRA, which are all published papers. We have also provided the parameter ratio in all tables.
>
> ## References
> **[1] Choucry, A., Nomoto, M., & Inokuchi, K. (2024). Engram mechanisms of memory linking and identity. Nature Reviews Neuroscience, 25(6), 375-392.**

---

### Official Review · Reviewer_WoLR · 2025-03-14

**Overall Recommendation:** 3

**Summary:**

The authors of this paper introduce a method called Synapse and Neuron (SAN), which decomposes and propagates scaling components from anterior feature adjustment vectors to posterior weight matrices. Extensive experimentation is performed by combining SAN with multiple PEFT strategies demonstrating the performance improvements achievable using SAN.

**Claims And Evidence:**

The core idea of explicit propagation of scaling components to subsequent layers by decomposing adjusting vectors from preceding layers  is interesting and its advantage is supported well by the experiments. However, the claim regarding LTP/LTD seems a bit stretched.

**Essential References Not Discussed:**

The paper discusses all major references.

**Experimental Designs Or Analyses:**

The authors present comprehensive experimental results that demonstrate the effectiveness of the proposed method across a wide range of vision, language, and vision-language tasks.

**Methods And Evaluation Criteria:**

Yes, the evaluation criteria used for underscoring the advantage of the proposed PEFT strategy seems sound.

**Other Comments Or Suggestions:**

The paper is easy to read.

**Other Strengths And Weaknesses:**

Strengths: This paper proposes a simple yet novel PEFT approach by involving explicit propagation of the scaling vectors of the current layer to the parameters of the subsequent layer. Experimental evidence suggest improved performance when combined with different PEFT strategies.

Weakness: The connection to LTP/LTD is unclear.

**Questions For Authors:**

1) For SAN+LoRA is the scaling factor propagated to the next layer that uses LoRA or just the immediate next layer of the model?

**Relation To Broader Scientific Literature:**

This article pertains to PEFT methodologies of LLMs, which is highly relevant for the broader ML and NLP community.

**Theoretical Claims:**

The paper provides some theoretical insights into how self-regularization is incorporated into the given framework.

---

> ### Author Rebuttal · Authors · 2025-03-29
>
> ## Response to Reviewer WoLR
> Thank you for your careful review and feedback. Your positive comments (such as praising our "sound evaluation criteria" and "easy to follow writing") have greatly support our paper. We have thoroughly reviewed your comments and address your key questions below:
> 1.  For SAN+LoRA is the scaling factor propagated to the next layer that uses LoRA or just the immediate next layer of the model?
>
> 2.  The claim regarding LTP/LTD seems a bit stretched.
>
> ## Point-by-Point Response:
> 1.  ### Regarding SAN+LoRA propagation mechanism
> In our experiments, we followed the default LoRA placement settings used in DoRA **[1]** and SSF **[2]**:
> -   For LLaMA/LLaVA models: LoRA was added to `["q_proj", "k_proj", "v_proj", "up_proj", "down_proj"]`
>
> -   For vision models: LoRA was added to all Linear layers (except the head)
>
> This naturally led to scaling factor propagation to the immediate next layer. However, your question raises an interesting point: when LoRA modules are sparsely added (e.g., every N block), is it better to propagate the scaling factor to:
> -   The immediate next layer (even without LoRA)
>
> -   The next layer that uses LoRA (potentially skipping several blocks)
>
> Due to time constraints, we conducted a simple test on ViT-CIFAR100 using only two LoRA modules (added to the qkv_attn layers in block 0 and block 11). The results are shown below:
> | Method | Linear Probing | LoRA-16 (no propagation) |   LoRA-16 (propagation to next layer) | LoRA-16 (propagation to next LoRA) |
> |--|--|--|--|--|
> | CIFAR100 | 88.74% | 90.82% | 91.03% | 90.38% |
>
> We found that when LoRA modules are distantly placed, long-distance propagation does not yield superior results. This aligns with our findings in Section 4.5 where long-distance propagation occurs when randomly apply. Conversely, propagating to related adjacent layers shows benefits, even when those layers don't have LoRA modules. We hypothesize that this shifts the LoRA module's learning objective from modeling only the current layer's parameter updates to also accounting for subsequent layers' updates, which enhances LoRA's learning capacity.
>
> 2.  ### Connection to LTP/LTD mechanisms
> Our method's connection to Long-Term Potentiation/Depression (LTP/D) is straightforward: we explicitly introduce the propagation mechanism. Existing PEFT methods like LoRA, DoRA, and SSF (as shown in Figure 3) typically only model the current layer's updates. Even though DoRA and SSF decompose scaling factors, they don't leverage propagation.
>
> In neuroscience, synaptic development is directly modulated by neuronal activation patterns. LTP and LTD exemplify this: high-frequency activation of presynaptic neurons induces LTP, strengthening the synaptic connection, while low-frequency stimulation induces LTD, weakening it **[3] [4]**. This aligns with Hebbian learning—the principle that “what fires together, wires together.”
>
> In our SAN approach, the explicit propagation of scaling factors mimics this biological process. Rather than treating each layer’s update in isolation, SAN transfers the scaling factor learned from one layer to the next. This is analogous to how the potentiation (or depression) of a synapse in a neural circuit influences the subsequent neurons, ensuring that plastic changes are distributed in a coordinated manner. For example, if a layer’s output is amplified (akin to LTP), the following layer would receives this amplified signal as a prior guidance and adjusts its weights accordingly. Moreover, by propagating scaling factors, our method effectively connects trainable modules throughout the model:
> -   Locally: we shift the learning focus from modeling only single-layer updates to considering groups of related layers.
>
> -   Globally: All trainable modules throughout the model become "interlocked," simplifying the adjustment range each module needs to cover (Figure 2) and achieving better performance
>
> This biologically inspired mechanism, therefore, not only provides a theoretical foundation for our approach but also delivers practical benefits in model transfer learning.
> ## Reference
> **[1] Liu, S. Y., Wang, C. Y., Yin, H., Molchanov, P., Wang, Y. C. F., Cheng, K. T., & Chen, M. H. (2024, July). Dora: Weight-decomposed low-rank adaptation. In** _**Forty-first International Conference on**_ _**Machine Learning**_**.**
>
> **[2] Lian, D., Zhou, D., Feng, J., & Wang, X. (2022). Scaling & shifting your features: A new baseline for efficient model** **tuning****.** _**Advances in Neural Information Processing Systems**_**,** _**35**_**, 109-123.**
>
> **[3] Malenka, R. C., & Bear, M. F. (2004).** _**LTP**_ _**and**_ _**LTD** **: An embarrassment of riches**_**.** **Neuron****, 44(1), 5–21.**
>
> **[4] Tonegawa, S., Liu, X., Ramirez, S., & Redondo, R. (2015).** _**Memory engram cells have come of age**_**. Neuron, 87(5), 918–931.**

---

### Decision · Program_Chairs · 2025-05-01

**Decision:**

Accept (poster)

**Comment:**

Concerns were raised that the code and hyperparameter details were not available and will only be released in the future. However, I do not find a strong enough reason for rejection and I would be willing to give the authors the benefit of the doubt, trusting that they will release all code and hyperparameters in the camera-ready version.
Overall, there was minimal but positive enthusiasm for this contribution.